# Mitochondrial dysfunction remodels one-carbon metabolism in human cells

Xiaoyan Robert Bao[1,2,3†], Shao-En Ong[3‡], Olga Goldberger[1], Jun Peng[1,3], Rohit Sharma[1], Dawn A Thompson[3], Scott B Vafai[1,3], Andrew G Cox[4], Eizo Marutani[5], Fumito Ichinose[5], Wolfram Goessling[3,4], Aviv Regev[3,6], Steven A Carr[3], Clary B Clish[3], Vamsi K Mootha[1,2,3*]

[1]Department of Molecular Biology, Howard Hughes Medical Institute , Massachusetts General Hospital, Boston, United States; [2]Department of Systems Biology, Harvard Medical School, Boston, United States; [3]Broad Institute of MIT and Harvard, Cambridge, United States; [4]Genetics Division, Brigham and Women's Hospital, Harvard Medical School, Boston, United States; [5]Department of Anesthesia, Critical Care, and Pain Medicine, Masaachusetts General Hospital, Boston, United States; [6]Department of Biology, Howard Hughes Medical Institute, Massachusetts Institute of Technology, Cambridge, United States

*For correspondence: vamsi@hms.harvard.edu

Present address: †Berkeley Lights, Inc., Emeryville, United States; ‡Department of Pharmacology, University of Washington School of Medicine, Seattle, United States

**Abstract** Mitochondrial dysfunction is associated with a spectrum of human disorders, ranging from rare, inborn errors of metabolism to common, age-associated diseases such as neurodegeneration. How these lesions give rise to diverse pathology is not well understood, partly because their proximal consequences have not been well-studied in mammalian cells. Here we provide two lines of evidence that mitochondrial respiratory chain dysfunction leads to alterations in one-carbon metabolism pathways. First, using hypothesis-generating metabolic, proteomic, and transcriptional profiling, followed by confirmatory experiments, we report that mitochondrial DNA depletion leads to an ATF4-mediated increase in serine biosynthesis and transsulfuration. Second, we show that lesioning the respiratory chain impairs mitochondrial production of formate from serine, and that in some cells, respiratory chain inhibition leads to growth defects upon serine withdrawal that are rescuable with purine or formate supplementation. Our work underscores the connection between the respiratory chain and one-carbon metabolism with implications for understanding mitochondrial pathogenesis.

## Introduction

Damaged mitochondrial respiratory chains play a key role in the pathogenesis of rare congenital metabolic disorders, as well as in a number of age-associated disorders such as diabetes (*Szendroedi et al., 2012*), neurodegenerative disease (*Schapira et al., 1989*), and aging. Mitochondrial respiratory chain components are encoded by two genomes; respiratory chain proteins encoded by mitochondrial DNA (mtDNA) are expressed in all tissues, yet inherited lesions within mtDNA can lead to varying tissue pathology (*Koopman et al., 2012*; *Vafai and Mootha, 2012*), suggesting a complex interplay between the primary respiratory chain dysfunction and the compensatory adaptations to that dysfunction. Improved understanding of cellular responses to respiratory chain dysfunction promises to deepen our understanding of mitochondrial disease pathogenesis, nominate new biomarkers (*Suomalainen et al., 2011*), and motivate new therapeutic strategies (*Zhang et al., 2013*).

Cellular responses to respiratory chain dysfunction, collectively known as the mitochondrial retrograde response, have been studied in a number of organisms (*Haynes et al., 2013*; *Liu and Butow,*

**eLife digest** Mitochondria are found within virtually all of our body's cells and are best known as their power plants. Damaged mitochondria cause many diseases in humans – from rare, inherited metabolic disorders that cause symptoms including muscle weakness and developmental problems, to age-related diseases such as diabetes and Parkinson's disease.

How does mitochondrial damage lead to such a variety of symptoms and conditions? To answer this question, researchers must understand how cells respond to and compensate for such damage.

To mimic mitochondrial failure, Bao et al. reduced the amount of DNA in the mitochondria of human cells and observed that this caused the cells to accumulate more of an amino acid called serine. Further investigation showed that this accumulation comes in part from cells producing more serine, and that a protein called Activating Transcription Factor 4 is responsible for increasing the expression of the genes needed to produce serine in the cells.

Bao et al. also found that damaged mitochondria are less able to consume serine to produce a compound called formate, which is a precursor for DNA building blocks. If cells cannot acquire enough extra serine to compensate for this inefficiency, they cannot produce some of the building blocks required to make DNA and other critical compounds in the cell. Supplementing the cells with formate or the DNA building blocks enabled the cells to recover, which suggests that formate supplements may help to treat some mitochondrial disorders.

At a higher level, these results suggest that the mitochondrion's role as a major chemical factory in the cell, and not just as the power plant, may also contribute to disease when the mitochondria are broken. Further work is now needed to investigate how cells know to turn on Activating Transcription Factor 4 when their mitochondria are damaged. It also remains to be discovered whether this reduces or exacerbates the symptoms of mitochondrial disease.

2006). In yeast, early application of genomic profiling and genetic studies identified a transcriptional program (*Liu and Butow, 2006*) that senses respiratory chain dysfunction, rewires metabolism to bypass a congested tricarboxylic acid (TCA) cycle, and promotes cellular survival. In worms, errors in mitochondrial biosynthesis trigger the mitochondrial unfolded protein response (UPR$^{mt}$) to activate transcription of mitochondrial proteases and chaperones, proteins that reduce oxidative stress, and glycolytic enzymes (*Nargund et al., 2012*). In flies, respiratory chain dysfunction signals to the nucleus and cytosol via reactive oxygen species (ROS) and altered energetics to JNK and AMPK, respectively, to place a checkpoint on cell division (*Owusu-Ansah et al., 2008*). Induction of a mitochondrial retrograde response in flies has recently been shown to suppress age-dependent degradation of mitochondria (*Owusu-Ansah et al., 2013*).

Studies of mammalian cells have identified a small collection of cellular responses to respiratory chain dysfunction. Classic studies showed that loss of mtDNA gives rise to uridine and pyruvate dependency (*King and Attardi, 1989*). Altered mitochondrial calcium uptake stemming from collapse of the mitochondrial membrane potential triggers metabolic alterations (*Amuthan et al., 2001*). Loss of the mtDNA induces expression of mitochondrial chaperones (*Martinus et al., 1996*). Recently, a mouse model of mitochondrial respiratory chain disease has been shown to mount an ATF4-mediated, starvation-like transcriptional response (*Tyynismaa et al., 2010*), though the functional relevance of this response remains unclear. At present, a systematic picture of how mammalian cells respond to respiratory chain dysfunction at the levels of both gene expression and metabolism is lacking; such a picture could help to unify the above observations, while also pointing to novel responsive and adaptive pathways.

We modeled mitochondrial respiratory chain dysfunction in human HEK-293 cells by depleting them of mtDNA. In human cells, mtDNA is required for expression of 13 structural subunits of the oxidative phosphorylation (OXPHOS) complexes that make up the respiratory chain; human patients with mtDNA depletion exhibit severe multicomplex respiratory chain deficiency (*Shoffner, 2005*). To generate hypotheses about pathways remodeling in respons to mitochondrial dysfunction, we systematically characterized the response of HEK-293 cells to mtDNA depletion using complementary metabolite, RNA, and protein profiling.

Integrated analysis of the profiles raised the hypothesis that one carbon metabolism, especially serine biosynthesis and transsulfuration, remodels in response to mitochondrial dysfunction. We confirmed this hypothesis with mechanistic follow-up experiments. We then explored alterations in serine metabolism in greater detail, and discovered that cells experience impairment of mitochondrial one-carbon synthesis upon respiratory chain dysfunction. In some cell lines, this leads to dependence on exogenously provided serine in the presence of mitochondrial dysfunction. Thus, our study identifies a new metabolic vulnerability of cells facing mitochondrial stress, with implications for our understanding and treatment of human mitochondrial disorders.

## Results

### A cellular model of mitochondrial dysfunction

We stably transfected T-REx-293 cells, a HEK-293-derived cell line expressing the tet repressor, with a plasmid that expresses a dominant-negative mutant of DNA polymerase gamma (POLGdn) (*Jazayeri et al., 2003*) under tet repressor control. Doxycycline-triggered expression of POLGdn halts replication of mtDNA, which is then diluted as cells continue to divide (*Figure 1a*). mtDNA-encoded OXPHOS complex components become depleted (*Figure 1a*) (*Jazayeri et al., 2003*), cellular respiratory capacity becomes compromised (*Figure 1—figure supplement 2*), and after several

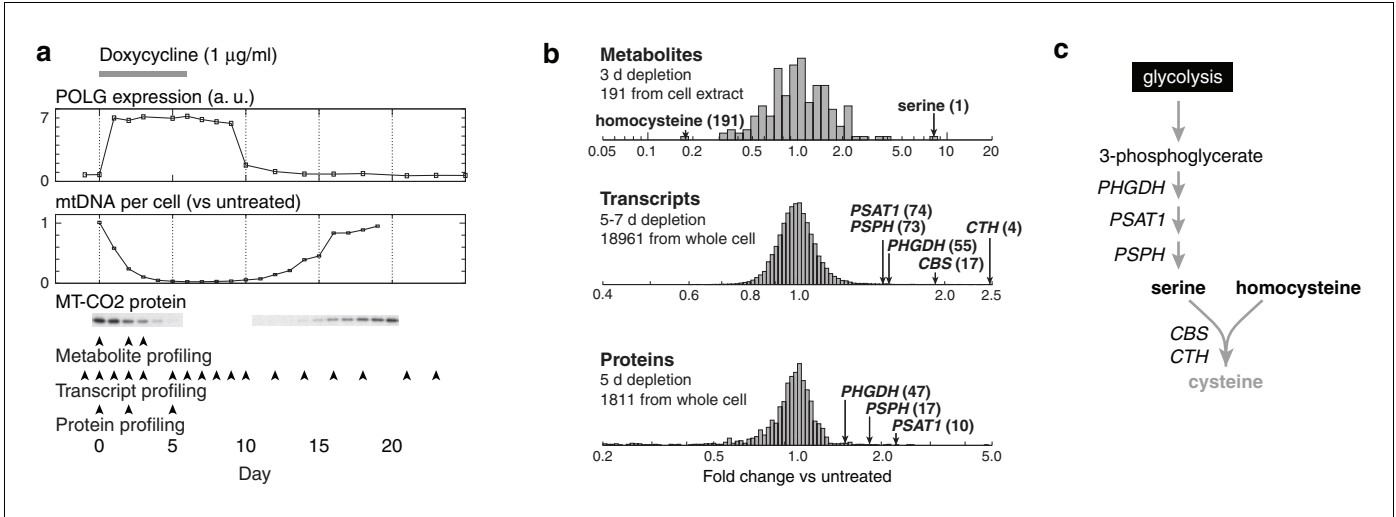

**Figure 1.** Integrated RNA, protein, and metabolite profiling of mtDNA depletion. (a) Experimental model. Doxycycline treatment induces PolGdn expression, mtDNA depletion, and reduction in oxidative phosphorylation complexes ($n = 1$ for all data). Arrowheads indicate time points analyzed by metabolite, proteomic, and transcriptional profiling. (b) Results from hypothesis-generating integrated profiling ($n = 2$ for metabolites; $n = 1$ for transcripts and proteins) showing serine- and homocysteine-related measurements. Numbers in parentheses represent ranks of the respective measurements. (c) Serine biosynthesis and transsulfuration pathways.

The following figure supplements are available for figure 1:

**Figure supplement 1.** mtDNA depletion time courses using PolGdn overexpression or EtBr treatment.

**Figure supplement 2.** Changes in oxygen consumption induced by mtDNA depletion.

**Figure supplement 3.** Metabolite profiling results.

**Figure supplement 4.** Spent media metabolite levels shown relative to levels in base media.

**Figure supplement 5.** Transcriptional profiling results.

**Figure supplement 6.** Protein profiling results ($n = 1$) at 2 d (yellow) and 5 d (red) of mtDNA depletion.

days, cell growth also slows. Removal of doxycycline allows recovery of mtDNA content and cell growth (*Figure 1—figure supplement 1*). Some of the recovery could be due to selection of cells that lose tet-inducible POLGdn expression (see Materials and methods), so we repeated these experiments using 100 ng/ml ethidium bromide treatment. Ethidium bromide directly inhibits mito-chondrial DNA replication, and gave similar results as POLGdn expression (*Figure 1—figure supplement 1*).

## Integrated profiling of cells during mtDNA depletion

To generate hypotheses as to what biochemical changes arise from POLGdn-induced mtDNA deple-tion, we performed initial studies using three profiling modalities. We performed metabolite profil-ing using targeted mass spectrometry (control and days 2 and 3; *Figure 1—figure supplement 3*; *Figure 1—figure supplement 4*; and *Supplementary file 1*), transcriptional profiling using microar-rays (selected days between 1 and 25, with two untreated controls; *Figure 1—figure supplement 5* and *Supplementary file 2*), and protein profiling using mass spectrometry (control and days 2 and 5; *Figure 1—figure supplement 6* and *Supplementary file 3*).

We stress that these experiments were performed with limited numbers of replicates: one tran-scriptional profiling replicate for each of 18 time points; one protein profiling replicate for each of 3 time points; and two metabolite profiling replicates for each of 3 time points and 2 different sample types. The paucity of replicates prevents us from drawing definitive conclusions from the individual profiles. Any hypotheses posed by analyzing these profiles must be considered preliminary and sub-ject to confirmation and validation.

When jointly analyzed, the profiling methodologies suggest increases in transsulfuration and ser-ine biosynthesis (*Figure 1b* and *Figure 1c*). Serine itself is the most strongly increased metabolite in all of our analyses, and the serine biosynthesis enzymes are among the most strongly upregulated proteins in our proteomic data. Transcriptional profiling reveals elevation of the entire serine biosyn-thetic pathway and two high-affinity human serine transporters (SLC1A4 and SLC1A5; Supplemen-tary Table 2a). While our metabolite profiling did not measure cysteine, we did find that the precursor for transsulfuration, homocysteine, was the most strongly decreased metabolite in all of our analysis. Further, we observed increased taurine, a known product of cysteine breakdown, and increased $\alpha$-hydroxybutyrate, a byproduct of transsulfuration, in metabolite profiling of spent media (*Figure 1—figure supplement 3*). Finally, we observed transcriptional activation of transsulfuration genes as well as SLC7A11, the dominant cystine transporter of the plasma membrane.

We next performed experiments to confirm the hypothesis that serine metabolism and transsulfu-ration are altered upon mtDNA depletion, and to understand the molecular basis of these alterations.

## Transcriptional changes arise from ATF4 activation

To discover *cis*-regulatory motifs and factors that might be responsible for our observed transcrip-tional changes, we performed motifADE analysis (*Mootha et al., 2004*) on our transcriptional pro-files. motifADE scores evolutionarily conserved *cis*-motifs for their relative enrichment within the vicinity of the transcription start sites of genes that are differentially expressed. Among all possible 6, 7, 8-mer, and gapped 9-mer motifs (*Figure 2a*, *Supplementary file 2*), the highest-scoring motif was the 8-mer 5'-TGATGCAA-3' ($p \approx 1.8 \times 10^4$, Mann-Whitney U test, adjusted for number of motifs tested), which is strikingly similar to consensus ATF4-C/EBP binding site TGATGHAAH (*Kilberg et al., 2009*). Indeed 26 of the 50 genes (52%) most upregulated in response to mtDNA depletion contain this consensus ATF4-C/EBP binding site near their transcription start sites (*Supplementary file 2*), representing a strong enrichment over the 14% seen across the genome ($p < 10^{-9}$, binomial test).

We confirmed ATF4 involvement in mtDNA depletion-elicited transcriptional changes by overex-pressing an N terminally truncated GADD34 protein (GADD34△N), which inhibits ATF4 activation (*Novoa et al., 2001*), in the parent T-REx-293 cell line. We tested four genes that were among the most highly up-regulated in our microarray dataset. We confirmed that these genes were also highly upregulated upon mtDNA depletion with EtBr, and that GADD34△N overexpression inhibits this upregulation (*Figure 2b*).

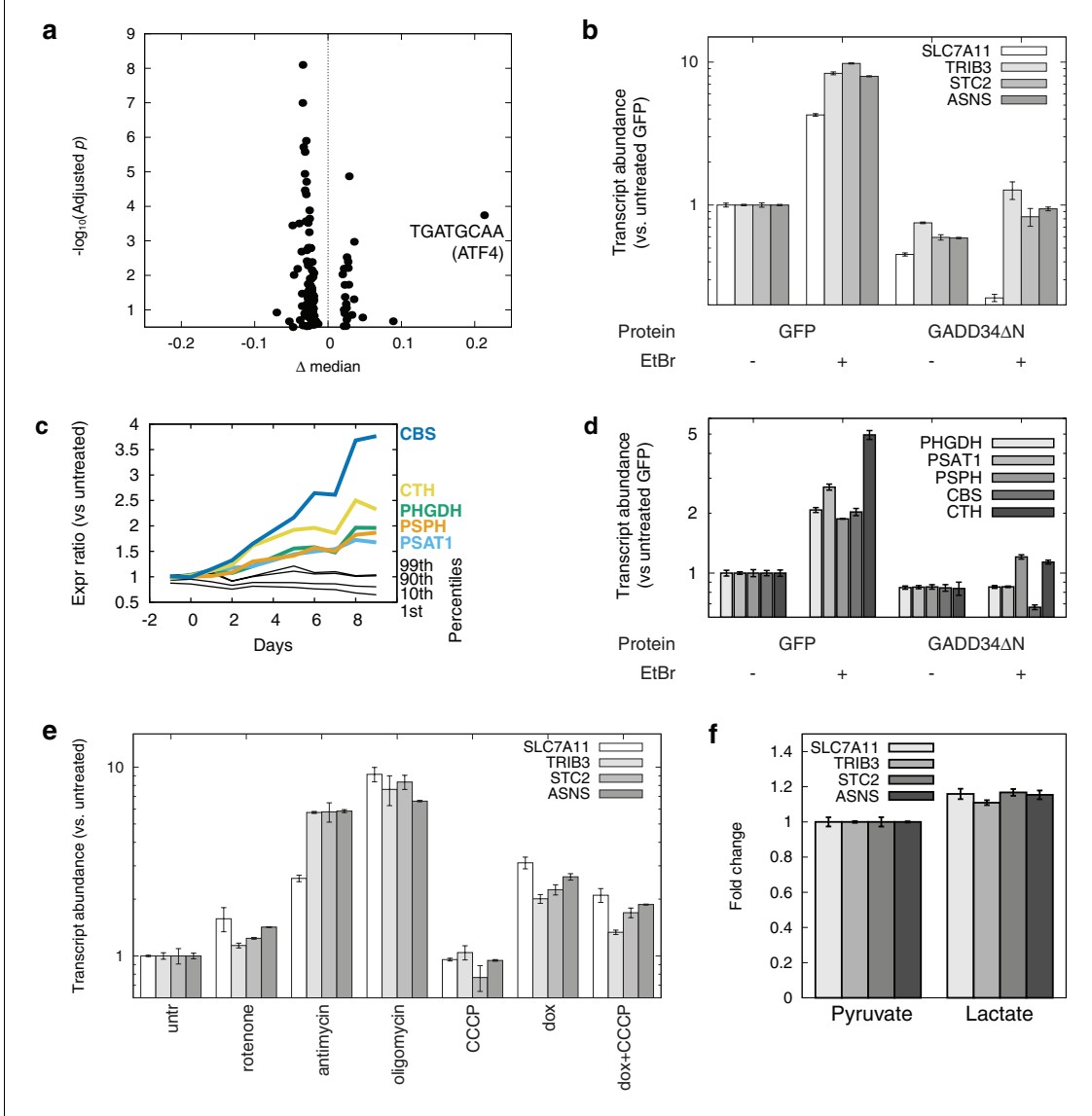

**Figure 2.** mtDNA depletion activates ATF4. (a) Volcano plot of motifADE analysis results (see Results). Δ median denotes the normalized rank of the median gene associated with each motif. (b) Fold changes of four of the most-upregulated genes in microarray data, in response to mtDNA depletion by ethidium bromide (EtBr) treatment (100 ng/ml, 9 d), and with expression of either GFP or GADD34ΔN. (c) Activation of serine and homocysteine biosynthesis genes, compared to that of genes at the 1st, 10th, 90th, and 99th percentiles in each transcriptional profiling timepoint. (d) Activation of serine and cysteine biosynthesis genes in response to mtDNA depletion, with and without GADD34ΔN expression. (e) Activation of ATF4 target genes in response to mitochondrial inhibitors. (f) Activation of ATF4 target genes in response to cytoplasmic redox imbanace elicited by lactate. $n$ = 3 for **b**, **d**, **e**, and **f**.

The following figure supplement is available for figure 2:

**Figure supplement 1.** Doxycycline control treatment data.

We performed a similar analysis of serine and cysteine biosynthesis genes. In the preliminary transcriptional profiling dataset, five of these genes were upregulated in nearly all the early mtDNA depletion time points (*Figure 2c*). GADD34△N overexpression blunted activation of all five genes in response to EtBr-induced mtDNA depletion (*Figure 2d*). ATF4 has previously been shown to activate expression of serine biosynthesis genes (*Ye et al., 2012*) and transsulfuration genes (*Dickhout et al., 2012*). Furthermore, chromatin immunoprecipitation experiments have shown

direct ATF4 protein binding to the promoter regions of PHGDH, PSAT1, and CTH (*Han et al., 2013*).

We used a series of mitochondrial inhibitors to determine what bioenergetic parameters might be upstream of ATF4 activation (*Figure 2e*). Inhibition of OXPHOS complexes III and V using antimycin and oligomycin, respectively, gives the strongest activation of genes shown above to be ATF4-responsive. Inhibition of complex I using rotenone gives subtle activation. Membrane potential dissipation using the uncoupler carbonyl cyanide *m*-chlorophenyl hydrazone (CCCP) fails to elicit activation of these genes, and indeed partially reversed ATF4 activation from mtDNA depletion. These results suggest that mtDNA depletion-triggered activation of ATF4 arises from redox stress due to a stalled respiratory chain, and not from defects in ATP synthesis or membrane depolarization.

To further investigate redox effects on ATF4 activation, we manipulated NAD+/NADH in cells by growing them in media with either 1 mM pyruvate or 1 mM lactate. Pyruvate and lactate are known to freely cross the plasma membrane, and altering the extracellular pyruvate:lactate ratio allows control over the cytoplasmic NAD+/NADH ratio through the action of lactate dehydrogenase (*Bücher et al., 1972*; *Williamson et al., 1967*). Cells grown in media for 24 hrs with 1 mM lactate showed significantly higher expression of ATF4 target genes than with 1 mM pyruvate (*Figure 2f*). However, the magnitude of the changes is modest compared to those seen with either antimycin treatment or with mtDNA depletion. Therefore, while altered cellular redox balance appears to contribute to ATF4 activation, other factors, such as oxidative stress from stalling of the respiratory chain, may also be important when mtDNA is depleted.

## Transcriptional and metabolic changes are independent of doxycycline toxicity

Because doxycycline can itself inhibit mitochondrial translation (*Ugalde et al., 2004*; *Wang et al., 2015*), we wondered whether direct doxycycline toxicity, and not doxycycline-induced POLGdn, was responsible for the transcriptional effects we observe in our profiling experiments. We used microarrays to measure the effect of 6 d doxycycline treatment on T-REx-293 cells that did not express POLGdn. ATF4 target genes showed much lower activation in doxycycline-treated T-REx-293 cells than in cells with POLGdn induction (*Figure 2—figure supplement 1a*). When we performed motifADE on the doxycycline control data and considered Bonferroni corrected *p*-values, the ATF4 motif TTGCATCA was not significant (*Figure 2—figure supplement 1b*).

## Increased H₂S production

Transsulfuration is important for generating $H_2S$, an emerging gaseous signaling molecule important for many aspects of mammalian physiology (*Kabil et al., 2014*). Given our transcriptional and metabolic profiles suggest an activation of transsulfuration, we wondered if mtDNA depletion could also increase $H_2S$ production.

To confirm that mtDNA depletion leads to decreased homocysteine abundance, we performed focused measurements of homocysteine levels in T-Rex-293 cells depleted of mtDNA using ethidium bromide (*Figure 3a*). Next, we used two different methods to detect $H_2S$. First, we reacted cell extracts with monobromobimane (MBB) and detected $H_2S$ adducts using HPLC (*Tokuda et al., 2012*). Second, we detected sulfane sulfur species in cell extracts using SSP4, a fluorescent probe (*Marutani et al., 2014*). $H_2S$ degradation requires a functioning respiratory chain (*Hildebrandt and Grieshaber, 2008*), so we used acute antimycin A treatment as a control for changes in $H_2S$ degradation rate. Both methods revealed increased $H_2S$ accumulation as a consequence of mtDNA depletion, greater than the amount induced by acute treatment with antimycin A (*Figure 3b*). These data support the notion that transcriptional changes elicited by mtDNA depletion have the effect of increasing cellular $H_2S$ production.

## Increased serine synthesis and decreased serine consumption

We confirmed alterations in serine using EtBr-treated T-Rex-293 cells, ruling out possible side effects due to doxycycline toxicity as well as statistical anomalies arising from the small sample size used in the initial metabolite profiling (*Figure 4a*).

In the initial metabolite profiling data, serine was the most strongly consumed metabolite, by fold change compared to base media, in untreated cells (*Figure 1—figure supplement 4*). Therefore,

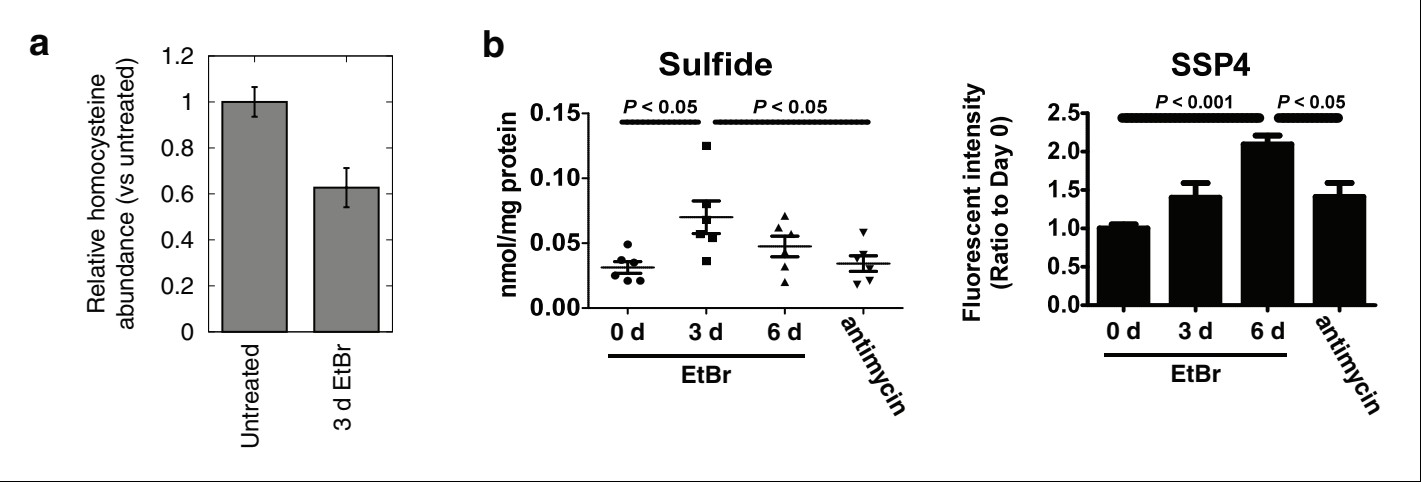

**Figure 3.** Alterations in transsulfuration-associated metabolites upon mtDNA depletion. (**a**) Confirmation of decreased homocysteine abundance in spent media with EtBr-induced mtDNA depletion. $n = 3$. (**b**) Hydrogen sulfide levels in cells measured either directly (sulfide) or indirectly by its sulfane products (SSP4). Acute antimycin treatment (1 hr) was used to control for increased $H_2S$ levels arising from decreased $H_2S$ degradation due to loss of the respiratory chain. $n = 6$ for both plots.

increased endpoint serine in the spent media atop mtDNA-depleted cells could reflect either increased synthesis or decreased consumption. To determine the contribution of serine biosynthesis to our observed increase in serine abundance, we performed isotopic tracer analysis using $^{13}C$ labeled glucose (***Chaneton et al., 2012***). To avoid product inhibition of serine biosynthesis (***Fell and Snell, 1988***), we used a low (50 µM) concentration of unlabeled serine in the labeled glucose media. The analysis (***Figure 4b***) shows that in response to mtDNA depletion, cells both produce more serine from glucose and take up less serine from the media. We estimate that the cell volume is at least 1000 fold smaller than the media volume in our experiments. Therefore, decreased serine uptake implies decreased overall serine consumption and not simply decreased serine accumulation in the cytoplasm.

## Respiratory chain dysfunction compromises mitochondrial formate production

To explain why serine consumption decreases, we considered the metabolic roles of serine in mammalian cells. In addition to its role in protein synthesis, serine is also a precursor for phospholipid biosynthesis and a major source of one-carbon (1C) units in folate metabolism, which supports purine and thymidylate synthesis as well as cellular methylation reactions (***Tibbetts and Appling, 2010***). Serine can allosterically activate PKM2 (***Chaneton et al., 2012***) and thereby regulate the exit of carbons from glycolysis (***Ye et al., 2012***). Serine is also a precursor for cysteine and glutathione biosynthesis via transsulfuration, and HCT116 cells lacking *p53* exhibit serine dependency that is rescuable with glutathione supplementation (***Maddocks et al., 2013***).

We became interested in the role of serine in 1C metabolism because mitochondria are the source of most of the 1C units used in cellular biosynthesis, and because consumption of serine for 1C metabolism involves an oxidation step that requires an NAD+ cofactor (***Tibbetts and Appling, 2010***). Thus, loss of NAD+ reoxidation due to the loss of the mitochondrial respiratory chain could conceivably lead to impairment of mitochondrial 1C metabolism, as theorized previously (***Desler et al., 2010***).

To test this model of 1C metabolism blockade in intact cells, we note that the NAD+-dependent step in mitochondrial metabolism is the oxidation of methylene-THF to formyl-THF by MTHFD2. Our model thus predicts an increase in abundance of methylene-THF. We inferred increased methylene-THF abundance by determining rates of serine isotopic scrambling (***Figure 4—figure supplement 1***). Briefly, $^{13}C_3$-labeled serine generates $^{13}C_2$-glycine and a $^{13}C$ labeled methylene-THF in mitochondria by the action of SHMT2. This labeled methylene-THF can recombine with an unlabeled glycine –

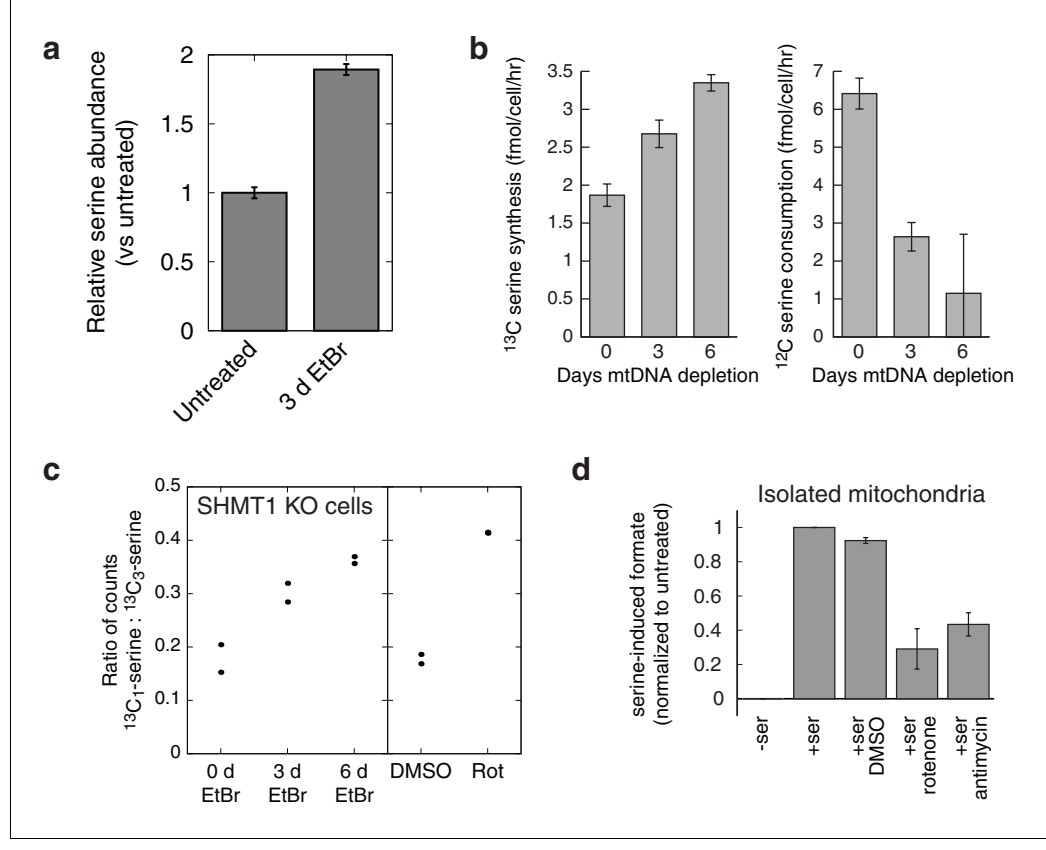

**Figure 4.** Respiratory chain dysfunction impairs mitochondrial 1C metabolism. (a) Confirmation of altered serine levels in spent media with EtBr-induced mtDNA depletion. $n = 3$. (b) Tracing serine metabolism using $^{13}C$-labeled glucose. Serine synthesis rates are inferred by labeling cells for 30 min with $^{13}C_6$-glucose and measuring the amount of $^{13}C_3$-serine that emerges. Simultaneously, serine consumption rates can be inferred by the amount of unlabeled serine that disappears during the labeling. $n = 3$. (c) Testing impairment of mitochondrial 1C metabolism using serine isotope scrambling in SHMT1 KO cells. Impairments in mitochondrial 1C metabolism downstream of $^{13}CH_2$-THF are reflected in increased generation of $^{13}C_1$-serine (see **Figure 4—figure supplement 1**). $n = 2$. (d) Testing impairment of mitochondrial 1C metabolism by assaying formate production from serine using isolated mitochondria with acute RC inhibition. $n = 3$.

The following figure supplements are available for figure 4:

**Figure supplement 1.** Rationale of serine isotope scrambling assay.

**Figure supplement 2.** Alterations in cellular $NAD^+/NADH$ ratio elicited by mitochondrial manipulations.

**Figure supplement 3.** Serine isotope scrambling in other cell types. *, different from 0 day EtBr with $p<0.05$ ($n = 3$).

**Figure supplement 4.** Determination of homocysteine remethylation.

which is present in excess – by the reverse action of SHMT2 to generate $^{13}C_1$-serine. The rate of emergence of $^{13}C_1$-serine thus reads out methylene THF abundance. We performed these experiments in SHMT1 knockout cells (see below) to avoid interference from cytoplasmic folate reactions, and found increased emergence of $^{13}C_1$-serine upon mtDNA depletion (**Figure 4c**). We observed similarly increased serine scrambling by treatment with rotenone, which directly inhibits NADH reoxidation through respiratory complex I. Measurement of cellular NAD+:NADH ratios confirms that these are altered with both mtDNA depletion and rotenone treatment (**Figure 4—figure**

*supplement 2*). Therefore, inhibition of mitochondrial respiration likely shifts the redox state of the mitochondrial folate pool in favor of reduced species such as methylene-THF.

To determine whether the shift in the mitochondrial redox state could inhibit mitochondrial 1C metabolism, we performed formate synthesis assays on isolated mitochondria treated with respiratory chain poisons. As shown in *Figure 4d*, both rotenone and antimycin A partially inhibit mitochondrial synthesis of formate from serine. García-Martínez and Appling performed similar experiments with mitochondria isolated from rat liver (*García-Martínez and Appling, 1993*), and also observed partial inhibition of formate synthesis with rotenone treatment.

## 1C units are not used for homocysteine remethylation in T-REx-293 cells

Elevations in homocysteine often signify insufficient one carbon pools. Hence, an important question is the degree to which mitochondrial 1C compromise is compatible with our observation of decreased homocysteine (*Figure 1b*). In mammals, homocysteine has two major fates: conversion to cysteine via transsulfuration, and conversion to methionine via remethylation; the latter process consumes cytoplasmic 1C units. Our hypothesized mitochondrial 1C compromise might therefore lead to increased homocysteine abundance and be in direct conflict with the strong observed decrease in homocysteine levels in both spent media and cell extract.

To examine this issue more closely, we performed deuterium tracer experiments to determine the extent of homocysteine remethylation in T-REx-293 cells. As shown in *Figure 4—figure supplement 4*, homocysteine remethylation using exogenously supplied, deuterated formate was nearly 1000-fold slower than synthesis of serine from deuterated formate. We conclude that remethylation is not a major fate for homocysteine in T-REx-293 cells, regardless of 1C availability. The observation of decreased homocysteine in our experiments, which we attribute to increased transsulfuraton, is therefore compatible with our model of mitochondrial 1C impairment.

## Mitochondrial dysfunction induces serine dependency in 293 cells

1C units are required for cellular biosynthesis, but treating T-REx-293 cells with either rotenone or antimycin, both of which impair mitochondrial 1C generation (*Figure 5a*), only gave subtle growth rates changes. To test the robustness of 1C metabolism in T-REx-293 cells undergoing mitochondrial dysfunction, we withdrew serine from cells treated with inhibitors. Antimycin- or rotenone-treated cells, but not control DMSO-treated cells, showed strongly impaired growth when serine was withdrawn (*Figure 5a*, *Figure 5—figure supplement 1a*). We observed a similar effect with serine withdrawal in cells depleted of mtDNA using EtBr (*Figure 5b*). A serine dose-response curve in the presence of antimycin (*Figure 5—figure supplement 1c*) shows optimal cell growth above about 100 µM serine in the media.

We note that this phenomenon is not a true auxotrophy, since cells are still able to synthesize serine. Indeed, if cells with mitochondrial dysfunction were truly serine auxotrophs, they would show increased serine consumption. Instead, we observed the opposite (*Figure 4b*). We surmise that these cells can overcome blockade of mitochondrial serine metabolism by mass action, but only at high cytoplasmic serine concentrations. Such a bypass mechanism would give rise to serine dependency if cells lose the cytoplasmic serine that they synthesize by diffusion to the media when exogenous serine is withheld, as has been observed for many cell types (*Eagle and Piez, 1962*).

We also observed serine dependency with oligomycin (*Figure 5—figure supplement 1a*), which others (*Maddocks et al., 2013*) have reported in HCT116 cells and attributed to serine involvement in energy metabolism. To distinguish the two models of serine dependency, we tested the effect of the mitochondrial uncoupler CCCP, which impairs energy production (*Figure 5—figure supplement 1b*) but increases NADH oxidation (*Figure 4—figure supplement 2*). CCCP did not give rise to serine dependency, and indeed a low dose of CCCP could rescue both the growth defect and the serine dependency arising from oligomycin treatment (*Figure 5—figure supplement 1a*). These data suggest that the mitochondrial dysfunction-induced serine dependency that we observe is related to the redox state of NAD cofactors, and not to energy impairment.

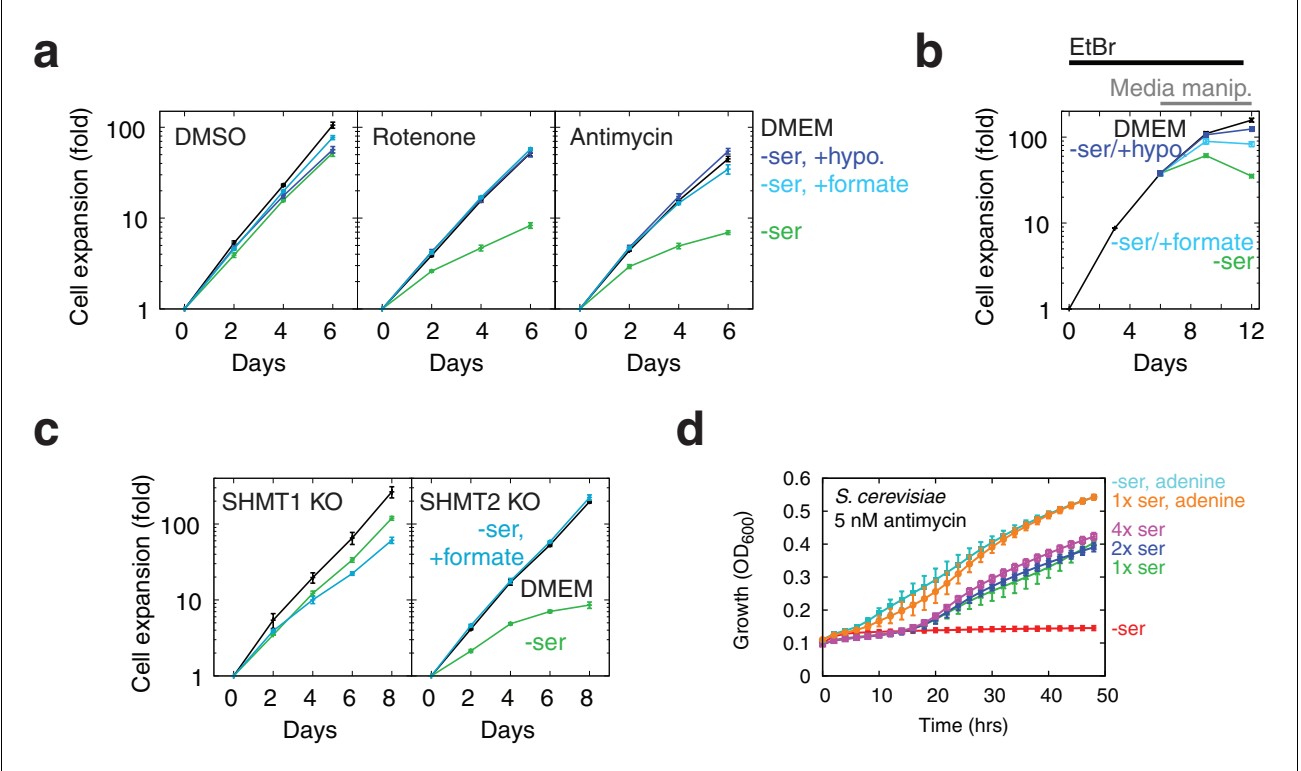

**Figure 5.** Serine dependence in cells with compromised mitochondrial function. (a) Growth of T-REx-293 cells treated with OXPHOS inhibitors, with and without serine, and with serine replaced by formate or hypoxanthine. (b) Same, but of T-REx-293 cells depleted of mtDNA using EtBr. (c) Growth of T-REx-293 cells with knockouts of SHMT1 or SHMT2, with and without serine, and with serine replaced by formate. (d) Growth of *S. cerevisiae* in nonfermentable media with 5 nM antimycin (see Materials and methods), in the presence or absence of serine and adenine. $n = 3$ for panels a-c; $n = 4$ for panel d.

The following figure supplements are available for figure 5:

**Figure supplement 1.** Additional data on RC inhibition-induced serine dependency in T-REx-293 cells.

**Figure supplement 2.** SHMT1 and SHMT2 single and double knockouts.

**Figure supplement 3.** RC inhibition-induced serine dependence in other cell types.

## Serine dependency can be rescued using 1C related metabolites

To further implicate 1C metabolism in serine dependency, we rescued this dependency using 1C related metabolites. Knockouts of MTHFD2 and MTHFD1L, two enzymes involved in mitochondrial formate synthesis, are embryonic lethal in mammals (*Di Pietro et al., 2002*; *Momb et al., 2013*), but some knockout phenotypes are rescuable with formate or hypoxanthine supplementation (*Momb et al., 2013*; *Patel et al., 2003*). Exogenous formate bypasses the need for its mitochondrial synthesis, and hypoxanthine is an alternate source of purines, the *de novo* synthesis of which requires 1C units. Experiments on OXPHOS-inhibited T-REx-293 cells with serine withdrawal showed near-complete rescue of growth when media was supplement with either 30 μM hypoxanthine or 3 mM sodium formate (*Figure 5a*). These concentrations of hypoxanthine and formate also gave strong rescue of growth defects in the absence of serine in mtDNA-depleted cells (*Figure 5b*). These data further support the notion that serine dependence in the face of mitochondrial dysfunction is related to 1C metabolism.

## Exploring compartment specificity using SHMT knockouts

Parallel folate pathways exist in the cytoplasm and the mitochondria (*Tibbetts and Appling, 2010*). To further determine the compartment specificity of serine fluxes, we used CRISPR/Cas9 (*Ran et al., 2013*) to knock out serine hydroxymethyltransferase (SHMT) genes in T-REx-293 cells (*Figure 5—figure supplement 2*). SHMT catalyzes the first step in folate-mediated production of formate from serine. Two isoforms exist in mammals: SHMT1, which is exclusively cytoplasmic, and SHMT2, which is mostly mitochondrial but has a cytoplasmic splice variant (*Anderson and Stover, 2009*). As expected, knockout of SHMT2, but not SHMT1, abolished formate synthesis from serine in isolated mitochondria (*Figure 5—figure supplement 2c*). In the presence of serine, neither SHMT1 KO nor SHMT2 KO cell lines required formate or hypoxanthine for growth, indicating that serine can be used to generate 1C units in either compartment. Without serine in the culturing media, SHMT2 KO cells showed strongly suppressed growth that could be rescued with formate, similarly to cells treated with RC inhibitors (*Figure 5c*), whereas SHMT1 KO cells did not. We hypothesize that high levels of exogenous serine are required to drive conversion of serine to formyl-THF in the cytoplasm via SHMT1, likely because MTHFD1, the cytoplasmic enzyme that makes formate, is coupled to NADPH and thus favors formyl-THF consumption rather than generation (*Tibbetts and Appling, 2010*). This provides a possible explanation for why high levels of serine can rescue 1C synthesis in T-REx-293 cells with compromised RC function.

Mammalian cells can also derive 1C units from breakdown of choline and glycine, which could in principle support growth in the absence of serine (*Tibbetts and Appling, 2010*). To investigate the roles of these other 1C precursors, we generated T-REx-293 cells with simultaneous knockout of both SHMT1 and SHMT2 genes. Supplementation with hypoxanthine and thymidine (HT) is required for cell growth in the absence of 1C metabolism (*Hakala, 1957*), since 1C units are used for the synthesis of purines and of thymidylate (*Tibbetts and Appling, 2010*). Therefore, we supplemented media with HT when we generated these double-KO (DKO) clones. DKO cells so generated were unable to grow on unsupplemented media, whereas most single knockouts and WT cells were able to grow (*Figure 5—figure supplement 2a*). In all cases, formate (3 mM) could replace HT to support DKO cell growth (not shown). We conclude from this that other sources of 1C are unable to completely replace serine-derived 1C units. Interestingly, hypoxanthine alone was also able to partially rescue DKO cell growth (*Figure 5—figure supplement 2e*), suggesting that an alternative pathway might be sufficient to supply 1C for thymidylate synthesis. Finally, DKO cells, when rescued using either formate or hypoxanthine, were no longer serine dependent even with antimycin treatment (*Figure 5—figure supplement 2f*), indicating that no other functions of serine account for its requirement in cells undergoing RC inhibition.

## Other cell types

We performed serine scrambling assays in five cell lines – T-Rex293, 293T, HeLa, C2C12, and MCH58 – to determine whether mtDNA depletion could induce 1C deficits in each of them. Instead of using SHMT1 knockout cells, we sought to suppress cytoplasmic SHMT activity by using lower concentrations of serine in the labeling media. Except with MCH58 cells, where high variance in the untreated samples prevented us from reaching any conclusions, we consistently observed increased ratios of $^{13}C_1$-serine to $^{13}C_3$-serine, with at least one time point showing a statistically significant increase for all the other cell lines tested (*Figure 4—figure supplement 3*).

However, impairment of 1C metabolism only led to serine dependence, manifesting as a proliferation defect upon serine withdrawal, in a small handful of cell lines: besides T-Rex293, we only observed similar results in 293MSR and U251 cells (*Figure 5—figure supplement 3*); a recent report (*Maddocks et al., 2013*) additionally shows serine dependence of HCT116 cells treated with oligomycin. We failed to observe serine dependence in many other cell lines, including 293T cells which, like T-Rex293 and 293MSR cells, are derived from HEK-293 cells. We surmise that expression of the large or small T antigen in 293T cells enables them to better cope with respiratory chain dysfunction-induced 1C compromise.

Although few mammalian cells demonstrate serine dependence in the presence of respiratory chain inhibition, we have observed similar phenomena in *S. cerevisiae* grown in nonfermentable media with sub-lethal doses of antimycin: serine withdrawal exacerbates antimycin-induced growth defects in a manner rescuable with purine supplementation (*Figure 5d*).

## Discussion

Classic inborn errors of metabolism are due to lesions within linear pathways, leading to accumulation of upstream substrates (that can be toxic) or depletion of essential downstream molecules. Therapeutic strategies for these diseases are often aimed at limiting substrate accumulation or replenishing of downstream factors. In respiratory chain dysfunction, however, the resultant metabolic derangements are manifold because of the numerous mitochondrial and cytoplasmic processes that are intimately coupled to the respiratory chain (*Vafai and Mootha, 2012*). Well-known metabolic alterations associated with mitochondrial disease include increased reliance on glycolysis as a source of ATP (*Robinson et al., 1992*), dependence on pyruvate to support aspartate biosynthesis (*Sullivan et al., 2015*; *Birsoy et al., 2015*; *Cardaci et al., 2015*), and inability to synthesize pyrimidines (*King and Attardi, 1989*).

In this paper, we offer two lines of evidence to support the idea that cellular 1C metabolism is also altered upon respiratory chain dysfunction. First, generating hypotheses with metabolic, proteomic, and transcriptional profiling and following these observations up with focused validation experiments, we report that cells activate serine biosynthesis and transsulfuration in response to mtDNA depletion. Second, we show in separate experiments that lesioning the respiratory chain impairs mitochondrial production of formate from serine. In a small number of cell types, this leads to growth defects upon serine withdrawal that is rescuable with purine or formate supplementation.

An important question is the degree to which our observations of one-carbon metabolism remodeling in proliferating cells are relevant to in vivo mitochondrial disease, which mostly affects post-proliferative tissue.

We note that our observation of ATF4 activation and alterations in serine metabolism and transsulfuration is richly supported by recent studies of mitochondrial disease. These have shown ATF4 activation by mitochondrial dysfunction in human cells lines (*Martínez-Reyes et al., 2012*; *Silva et al., 2009*), rodent models (*Tyynismaa et al., 2010*), and humans (*Crimi et al., 2005*). Increased transsulfuration has been observed following ATF4 activation in general (*Dickhout et al., 2012*), and mitochondrial dysfunction in particular (*Krug et al., 2014*). One of the byproducts of transsulfuration, α-hydroxybutyrate, has been recently identified as a promising biomarker for mitochondrial disease in humans (*Thompson Legault et al., 2015*) and mice (*Jain et al., 2016*). ATF4 is known to activate serine biosynthesis (*Ye et al., 2012*), and increased serine has been observed in a number of mitochondrial disease settings, albeit with less consistency. Muscle from male mice (but not female ones) with mitochondrial myopathy show increased serine levels (*Tyynismaa et al., 2010*). Serine was found to be highly elevated in urine from mitochondrial disease patients (*Smuts et al., 2013*) and proposed as a component of a novel biosignature for mitochondrial disease, but not found to be increased in the blood of a different cohort of mitochondrial disease patients (*Clarke et al., 2013*). Serine levels are increased in a mitochondrial Parkinson's disease model in *Drosophila* (*Tufi et al., 2014*), but not in *C. elegans* genetic models of mitochondrial disease (*Falk et al., 2008*; *Morgan et al., 2015*).

Furthermore, while serine dependency appears to be cell type specific, 1C compromise as evidenced by altered serine scrambling appears to be more general. We note that a recent report on the effect of mitochondrial inhibitors showed impaired serine uptake and formate release in differentiated C2C12 myotubes (*Xu et al., 2011*). Therefore, 1C compromise does not appear to be limited to proliferating cells. More recently, studies of mouse mitochondrial disease models have also noted imbalanced nucleotide pools and altered 1C-related metabolites (*Nikkanen et al., 2016*). Taken together, these data suggest that 1C metabolism is more efficient in the presence of a functioning respiratory chain in a variety of in vivo and in vitro contexts, and argues that the 1C compromise seen with respiratory chain dysfunction might be generalizable to whole body metabolism.

Whether the ATF4 response is adaptive in this context has not been clear. Our study shows ATF4-dependent activation of serine synthesis (*Ye et al., 2012*) following respiratory chain blockade, and poses the hypothesis that the serine synthesis-activating aspect of ATF4 might be adaptive by helping maintain cellular 1C availability. Indeed, ATF4 has recently been shown to mediate purine synthesis induction by mTORC1 (*Ben-Sahra et al., 2016*). Transsulfuration constitutes another potentially adaptive element of the ATF4 response program, since it supports synthesis of glutathione, a key cellular ROS scavenger. However, we have so far failed to observe deleterious effects of ablating the ATF4 response in T-REx-293 cells. Since the ATF4 integrated stress response program

encompasses a number of different genes and functions, we hypothesize that it contains both adaptive and maladaptive components. In support of this notion, a recent paper showed that inhibition of ATF4 activation is protective in rodent models of intracerebral hemorrhage (*Karuppagounder et al., 2016*).

Our work may provide mechanistic insights into the folate deficiency in mitochondrial disease, with therapeutic implications. Mitochondrial disorders are occasionally associated with cerebral folate deficiency (CFD), and some of these patients are known to respond to treatment with folinic acid (*Garcia-Cazorla et al., 2008*). No proven mechanism yet connects mitochondrial disease to CFD, but given that brain expresses very little SHMT1 (*Girgis et al., 1998*), our results pose the hypothesis that CFD may result from impaired mitochondrial formate synthesis (*Figure 4*). Formate supplementation has been shown to ameliorate defects associated with 1C gene knockout (*Momb et al., 2013*), and a recent report demonstrated that the bioenergetics defects of a fly model of Parkinson's disease could be reversed with supplementation of nucleotides (*Tufi et al., 2014*). Mitochondrial disease patients are commonly given folic acid as part of the 'mito cocktail,' with no proven efficacy. Our results suggests that these patients might lack the 1C units that are bound by the folate cofactors, and not the co-factors themselves. Our work raises the hypothesis that formate or nucleotide supplementation may be of benefit in some of respiratory chain diseases.

One of the most puzzling characteristics of mitochondrial diseases is their phenotypic heterogeneity and tissue-specific pathology (*Vafai and Mootha, 2012*). Our identification of 1C metabolism and transsulfuration as metabolic alterations associated with mitochondrial disease presents a new set of hypotheses that could help explain this heterogeneity. Different organs in the body differ in which 1C donors they utilize, in their ability to metabolize those 1C units (*Yoshida and Kikuchi, 1973*), and in their transsulfuration activity (*Mudd et al., 1965*). Because many of these one-carbon pathways are coupled to the respiratory chain, a lesion within the mitochondrion could in principle ripple out to 1C defects that could manifest in many different ways depending on the tissue and cell state. Rapidly proliferating cells might be unable to replicate their DNA due to nucleotide imbalances, whereas metabolically active cells in the liver and kidney might be deprived of essential purine-containing cofactors such as NAD+. Tissue-specific variations in 1C metabolism may contribute to the remarkable phenotypic variation of mitochondrial disease.

## Materials and methods

### Cell culture

T-REx-293 cells were obtained from Thermo Fisher (Waltham, MA). POLGdn cells were generated as in *Jazayeri et al. (2003)*. U251 cells were obtained from the National Cancer Institute. 293T, HeLa, and C2C12 were obtained from ATCC. MCH58 cells were a generous gift from Eric Shoubridge. We did not authenticate cell lines. However, we were able to differentiate C2C12 myoblasts into myotubes, and were able to PCR the SV40 small and large T antigens from 293T cell genomic DNA. All cells were cultured with DMEM (11995, Thermo Fisher) supplemented with 10% FBS (F6178, Sigma-Aldrich, St. Louis, MO) and grown at 37°C with 5% $CO_2$. DMEM without serine or glycine, but with 1 mM pyruvate and in every other respect identical to the DMEM obtained from Thermo Fisher, was custom-made (U.S. Biological, Salem, MA) and supplemented with 10% dialyzed serum (F0392, Sigma-Aldrich). All cell cultures were tested for mycoplasma contamination monthly and confirmed to be free of mycoplasma.

### mtDNA analysis

DNA was extracted from cells and analyzed by multiplex real time quantitative PCR as in *Baughman (2011)*. Briefly, samples of $5 \times 10^4$ cells were pelleted, aspirated, and suspended in 50 µl lysis buffer (25 mM NaOH, 0.2 mM EDTA). Cell lysate was heated to 95°C for 15 min to hydrolyze protein and RNA, and neutralized with addition of 50 µl neutralization buffer (40 mM Tris-HCl). 5 µl of 1:50 diluted neutralized cell lysate was analyzed using multiplexed TaqMan real-time quantitative PCR in 20 µl reactions, with a custom-synthesized assay for the AluYb8 repeat element to quantitate nuclear DNA copy number and a custom-synthesized assay for MT-ND2 to quantitate mitochondrial DNA copy number. The custom-synthesized AluYb8 Taqman assay consisted of the forward primer 5'-CTTGCAGTGAGCCGAGATT-3', the reverse primer 5'-GAGACGGAGTC-TCGCTCTGTC-3', and the

probe 5'-VIC-ACTGCAGTCCGCAGTCCGGCCT-MGBNFQ-3'. The custom-synthesized MT-ND2 assay consisted of the forward primer 5'-TGTTGGTTATACCCTTCCCGTACTA-3', the reverse primer 5'-CCTGCAAAG-ATGGTAGAGTAGATGA-3', and the probe 5'-6FAM-CCCTGGCCCAACCC-MGBNFQ-3'. To calibrate MT-ND2 PCR Ct values, we used dilution ladders of a chemically synthe-sized primer corresponding to the target of the quantitative PCR assay. To calibrate AluYb8 PCR Ct values, we used dilution ladders of total DNA extracted from human cells.

## mtDNA depletion growth curves and RNA extraction

PolGdn cells were split every 3 d. On each split, two 10 cm plates were seeded with 2.0, 1.0, and 0.5 M each for RNA and protein collection after 1, 2, and 3 d. RNA was extracted from cell plates using RNeasy columns (Qiagen, Germany) with DNase treatment to remove DNA contamination. For protein collection, cells were first trypsinized, suspended, and counted. Counts derived from protein collection were used for growth curve computation. $5 \times 10^4$ cell were used for mtDNA analysis. The remaining cells were then pelleted and lysed using RIPA buffer with cOmplete protease inhibitor (Roche, Switzerland). Ethidium bromide growth curves were generated in similar fashion, without plates for RNA samples. We note that continuous doxycycline treatment also gave rise to mtDNA repletion, albeit more slowly than when doxycycline was removed, raising the possibility that the strong selective pressure against POLGdn expression could yield mutant cells that lost POLGdn expression.

## Oxygen consumption measurements

Oxygen consumption measurements were performed in a Seahorse XF24 Analyzer (Agilent, Santa Clara, CA). Untreated and mtDNA-depleted POLGdn-expressing cells, suspended by trypsinization, were seeded at $10^5$ per well into Seahorse cell plates pre-treated with Cell-Tak (Becton Dickinson, Santa Clara, CA) according to manufacturer's recommendations. Cells were incubated in DMEM in Seahorse cell plates for 1 hr before oxygen consumption measurement.

## Metabolite profiling

PolGdn-expressing cells were seeded to twelve 6-cm plates at $5 \times 10^5$ ea in 3 ml DMEM + 10% FBS. Four of these plates had 1 μg/ml doxycycline, and another four had 1 μg/ml doxycycline added 1 d after seeding. 3 d after seeding, media was collected from plates. Plates were washed once with ice-cold PBS and aspirated. Polar metabolites were extracted from two plates of each condition using 1 ml -80°C 80% methanol:20% water. Nonpolar metabolites were extracted from two plates of each condition using 1 ml -80°C isopropanol.

A combination of three liquid chromatography tandem mass spectrometry (LC-MS) methoda was used to measure metabolites in cell extracts and spent media as described previously (*Townsend et al., 2013*). Briefly, polar metabolites were profiled using a 4000 QTRAP triple quadru-pole mass spectrometer (SCIEX; Framingham, MA) coupled to a 1200 Series pump (Agilent Technol-ogies; Santa Clara, CA) and an HTS PAL autosampler (Leap Technologies; Carrboro, NC). Media samples (10 μL) were extracted using 90 μL of 74.9:24.9:0.2 v/v/v acetonitrile/methanol/formic acid containing stable isotope-labeled internal standards (valine-d8, Isotec; and phenylalanine-d8, Cam-bridge Isotope Laboratories; Andover, MA) and centrifuged (10 min, 9000 x g, 4°C). Cell extracts and media extraction supernatants (10 μL) were injected directly onto a 150 x 2 mm Atlantis HILIC column (Waters; Milford, MA). The column was eluted isocratically at a flow rate of 250 μL/min with 5% mobile phase A (10 mM ammonium formate and 0.1% formic acid in water) for 1 min followed by a linear gradient to 40% mobile phase B (acetonitrile with 0.1% formic acid) over 10 min. MS anal-yses were carried out using electrospray ionization and selective multiple reaction monitoring scans in the positive ion mode. Declustering potentials and collision energies were optimized for each metabolite by infusion of reference standards before sample analyses. The ion spray voltage was 4.5 kV and the source temperature was 450°C. Polar metabolite were profiling in the negative ion mode using a 5500 QTRAP triple quadrupole mass spectrometer (SCIEX; Framingham, MA) coupled to an ACQUITY UPLC (Waters; Milford, MA). Media samples (30 μL) were extracted using 120 μL of 80% methanol containing inosine-$^{15}$N4, thymine-d4 and glycocholate-d4 internal standards (Cambridge Isotope Laboratories; Andover, MA) and were centrifuged (10 min, 9000 x g, 4°C). Cell and media extract supernatents (10 μL) were injected directly onto a 150 x 2.0 mm Luna NH2 column

(Phenomenex; Torrance, CA) that was eluted at a flow rate of 400 µL/min with initial conditions of 10% mobile phase A (20 mM ammonium acetate and 20 mM ammonium hydroxide in water) and 90% mobile phase B (10 mM ammonium hydroxide in 75:25 v/v acetonitrile/methanol) followed by a 10 min linear gradient to 100% mobile phase A. The ion spray voltage was -4.5 kV and the source temperature was 500°C. Lipid profiling was performed in the positive ion mode using a 4000 QTRAP triple quadrupole mass spectrometer (SCIEX; Framingham, MA) coupled to a 1100 Series pump (Agilent Technologies; Santa Clara, CA) and an HTS PAL autosampler (Leap Technologies; Carrboro, NC). Lipids were extracted from media (10 µL) using 190 µL of isopropanol containing 1-dodeca-noyl-2-tridecanoyl-sn-glycero-3-phosphocholine (Avanti Polar Lipids; Alabaster, AL). Cell and media extracts (10 µL) were injected directly onto a 150 x 3.0 mm Prosphere HP C4 column (Grace, Columbia, MD). The column was eluted isocratically with 80% mobile phase A (95:5:0.1 vol/vol/vol 10 mM ammonium acetate/methanol/acetic acid) for 2 min followed by a linear gradient to 80% mobile-phase B (99.9:0.1 vol/vol methanol/acetic acid) over 1 min, a linear gradient to 100% mobile phase B over 12 min, then 10 min at 100% mobile-phase B. MS analyses were carried out using electrospray ionization and Q1 scans in the positive ion mode. Ion spray voltage was 5.0 kV and source temperature was 400°C. For each lipid analyte, the first number denotes the total number of carbons in the lipid acyl chain(s) and the second number (after the colon) denotes the total number of double bonds in the lipid acyl chain(s).

For each method, internal standard peak areas were monitored for quality control. MultiQuant software (Version 1.1; AB SCIEX; Foster City, CA) was used for automated peak integration and metabolite peaks were manually reviewed for quality of integration and compared against a known standard to confirm identity.

Cell extract metabolite abundances were normalized to sample geometric means to adjust for cell number differences. We used a multi-step protocol to adjust spent media metabolite abundances. First, we measured a media volume change of 12.0% in 6 cm plates with 3 ml media over the course of 3 d, and re-scaled media metabolite measurements accordingly. Then, we used the log mean cell extract metabolite differences between the different samples to estimate the per-day growth rate difference due to doxycycline treatment. We then used the growth rate difference to estimate changes in the area-under-the-curve (AUC) exposure of media to cells and adjusted accordingly. While this adjustment entailed straightforward flux multiplication in the case of metabolites released to the media, metabolites strongly absorbed from the media were more difficult to treat because multiplication would give rise to cells absorbing more metabolite than was present in the base media. We therefore used an ad-hoc adjustment function for uptaken metabolites that satisfied three criteria: (1) its value and slope would match that for released metabolites at the zero flux boundary case; (2) it would be continuous for all uptake fluxes; (3) measurements of near-100% uptake would adjust to yield near-100% uptake. Our final adjustment function was

$$R_{adj} = \begin{cases} 1 + \alpha(R-1), & R>1 \\ \frac{R}{\alpha - R(\alpha-1)}, & 0<R<1 \end{cases}$$

where R is the evaporation-adjusted abundance ratio compared to base media, $\alpha$ the AUC adjustment factor, and $R_{adj}$ the growth-adjusted metabolite abundance ratio used in *Figure 1—figure supplement 3* and *Figure 1—figure supplement 4*.

## RNA profiling

RNA from 18 time points (*n* = 1 for each time point) were analyzed using Affymetrix (Santa Clara, CA) Human Genome U133 Plus 2.0 arrays. RNA sample processing, hybridization to Affymetrix U133 Plus 2.0 microarrays, and microarray imaging were performed according to manufacturer's recommendations. Raw microarray data were analyzed using the RMA algorithm with manufacturer-provided probeset definitions. Probesets were filtered in an ad-hoc manner according to three criteria: (a) probeset standard deviation across all the timepoint samples were required to be at least 7.5% of the probeset mean, to remove unchanging probesets; (b) maximum values for probesets were required to be at least 50, to remove probesets with insufficient signal; (c) total power in the second through sixth coefficients of the Fourier transform of the data were required to be at least as strong as the total power in the seventh through twelfth, to remove probesets with high frequency noise that we considered unlikely to be informative. Probesets that did not survive these filters were not

considered for any downstream analysis. To estimate the number of transcripts with significant changes at day 6, we pooled data from days 5–7 and compared these to data from one and zero days prior to initiation of doxycycline treatment.

Full microarray data have been deposited to GEO: http://www.ncbi.nlm.nih.gov/geo/query/acc.cgi?acc=GSE55311

## motifADE analysis

(*Mootha et al., 2004*) was performed using publicly available code (Broad Institute). Genes were ordered by the ratio of the average value from days 1 through 10 of dox treatment, to the average of untreated samples. Sequences flanking transcription start sites, with repeat sequences masked, were obtained from the UCSC genome browser. Mouse-human homology was used to improve detection of conserved transcription factor binding sites. We looked for all ungapped motifs between 6 and 8 in length, as well as 9mer motifs with single gaps. We observed the ATF4 signal regardless of whether we looked for bidirectional occurrence of the motifs or considered each direction separately; the results shown in *Figure 2a* and *Supplementary file 2* are for bidirectional searching.

## Protein overexpression

The N-terminal truncation of GADD34 reported in *Novoa et al. (2001)* was from CHO-K1 cells; we wished to obtain the analogous human version of the protein. We BLASTed the C. griseus GADD34 protein sequence against the human, and determined that the truncation point corresponded to position 308 in the H. sapiens GADD34 protein sequence (NCBI reference sequence symbol PPP1R15A). We amplified this fragment from T-REx-293-derived complementary DNA using the PCR primers GATGAAGAGGAGGGTGAGGTCAAG and GCCACGCCTCCCACTGAGG, performed an additional 30 cycles of PCR using Gateway-flanked primers that also incorporated an ATG start codon, and recombined the resulting PCR product into the pDONR221 Gateway entry vector using BP Clonase II (Thermo Fisher). We call the protein product of this insert GADD34△N. We encountered difficulties in selecting for lentivirally-transduced GADD34△N expression using puromycin resistance, and suspected that inhibition of the ATF4 response might make cells more susceptible to puromycin treatment. To work around this, we used restriction digestion to remove the puromycin resistance open reading frame from the pLX302 lentiviral expression vector (Addgene, Cambridge, MA) and used Gibson isothermal assembly (*Gibson et al., 2009*) to replace it with GFP. We recombined the GADD34△N insert into this new vector (which we call pLX-GFP), made virus with psPAX2 (Addgene) and pCMV-VSVG (Broad Institute) helper plasmids, and infected T-Rex293 cells at MOI < 1. We then selected for GFP expression with two rounds of fluorescence-activated cell sorting on a FACSAria II (Becton Dickinson). Control infections were done similarly, except with an additional copy of GFP instead of GADD34△N inserted using Gateway recombination.

## Protein profiling

We characterized protein changes (*n* = 1) for two different time points, plus a control. PolGdn cells were grown in SILAC labeled DMEM (Caisson Labs, Logan, UT) with 10% dialyzed FBS (Sigma). They were first seeded at $4.0 \times 10^6$ each into three sets of replicate 15-cm plates with heavy (R10K8), medium (R6K4), and light (R0K0) SILAC media and grown for 3 d. They were then split into eight 15-cm plates for each condition. Doxycycline was added at 1 µg/ml 2 d before the split to heavy labeled cells, and 1 d after the split to medium labeled cells. 3 d after the split, cells were trypsinized, and cells from replicate plates pooled and counted. $7 \times 10^6$ cells from each condition were mixed, pelleted, aspirated, suspended in 8 M urea and sonicated. This sample was treated with dithiothreitol (1 mM final), and iodoacetamide (6 mM final) and proteins fractionated by SDS-PAGE. Each gel lane was cut into ten slices of approximately equal staining intensity, in-gel digested with trypsin, and analyzed by liquid chromatography-mass spectrometry on a LTQ-Orbitrap (Thermo, Inc.) following the procedure in *Sancak et al. (2013)*. SILAC data were analyzed using MaxQuant ver.1.0.13.13 according to the procedure described in *Cox et al. (2009)*. A minimum of two quantified peptides was required for each quantified protein. The FDR for protein and peptide identification was set at 0.01. The IPI human database ver 3.65 was used and supplemental sequences from

mitochondrial DNA and common contaminants such as keratins and serum proteins were included at search time. Raw data are publicly available at www.broadinstitute.org/proteomics

## H$_2$S assays

Sulfide (sum of H$_2$S, HS$^-$, and S$^{2-}$) levels in the T-REx-293 cells were measured using monobromobimane (MBB)-based high performance liquid chromatography (HPLC) analysis as reported previously (*Marutani et al., 2012*; *Tokuda et al., 2012*). Briefly, cells were washed with ice-cold Tris-HCl (100 mM, pH 9.5, 0.1 mM DTPA) buffer, scraped, transfered to an eppendorf tube, and centrifuged to obtain the supernatant. MBB (10 mM in acetonitrile, 50 µl) was added to 100 µl of supernatant. After 30 min of incubation at room temperature in dark, 50 µl of 200 mM 5-sulfosalicylic acid (SSA) was added. After centrifugation, supernatant was analyzed by HPLC equipped with Agilent HPLC column C18 and Waters 2475 Multi λ fluorescence detector.

Sulfane sulfur levels in T-REx-293 cells were measured using a sulfane sulfur-specific fluorescent probe SSP4 (generous gift from Dr. Ming Xian, Washington State University) as reported previously (*Marutani et al., 2014*). Briefly, cells were washed with ice-cold Tris-HCl (100 mM, pH 9.5, 0.1 mM DTPA) buffer, scraped, transfered to an eppendorf tube, and centrifuged to obtain the supernatant. The supernatant was transferred into a 96 well-plate, incubated with SSP4 at 20 µM at 37°C for 30 min, and fluorescent intensity was read by a microplate reader (SpectraMax M5 Microplate reader, Molecular Devices, Sunnyvalem CA).

## $^{13}$C labeling to determine serine synthesis

PolGdn cells, treated for 0, 3, and 6 d with doxycycline to induce mtDNA depletion, were in triplicate 35 mm plates at approximately $2 \times 10^6$ cells per plate. Each plate was washed with warm PBS and aspirated. To each plate was then added 1 ml of warm $^{13}$C glucose media, which was glucose- and serine-free DMEM + 10% dialyzed FBS, supplemented with 25 mM [U-$^{13}$C]glucose (Cambridge Isotope Labs) and 50 µM unlabeled serine (Sigma). We used this lower concentration of serine (DMEM normally contains 400 µM) to avoid potential product-inhibition of serine synthesis (*Fell and Snell, 1988*). Cells were incubated with labeling media at 37°C for 30 min, after which labeling media was removed, pelleted at 2000 g for 20 s, and supernatant taken for LC-MS/MS quantitaton (see below). The cell pellet and cells on plates were trypsinized, pooled, resuspended, and counted to adjust serine fluxes for cell number.

## LC-MS/MS quantitation of $^{12}$C and $^{13}$C serine in media

Cell medium samples (previous paragraph) were prepared using nine volumes (1:9 v/v) of extraction solution containing 75% acetonitrile, 25% methanol, and 0.2% formic acid, and vortexed. Samples were then centrifuged at 11,000 rpm at 4°C for 8 min, and the supernatant was injected into LC-MS/MS. A series of $^{12}$C serine (BioUltra 99.5%, Sigma) and $^{13}$C serine ($^{13}$C$_3$, 99%, Cambridge Isotope Labs) standard solutions at concentrations of 0.1, 0.5, 1, 5, 10, 50 µM were prepared in serine-free base medium to generate a calibration curve. An Agilent 1260 HPLC system coupled with Q Exactive (Thermo Fisher) was used perform LC-MS/MS quantitation. LC Column was Atlantis HILIC Silica 2.1 × 150 mm column and particle size was 3 µm. Mobile Phase A was 0.1% formic acid, 10 mM ammonium formate. Mobile Phase B was 0.1% formic acid in acetonitrile. Column was at room temperature. 10 uL volume of samples was injected into LC-MS/MS. The flow rate was 0.25 mL/min. The LC gradient was as follows: 95% B was held from 0 to 0.5 min, then from 0.5 to 10.5 min, 95% B was linearly changed to 40%. From 10.5 to 15 min, 40% B was held. From 15 min to 17 min 40% B was changed to 95%. From 17 to 32 min, 95% B was held to equilibrate the column. Serine was quantified using the targeted MS2 method. For $^{12}$C serine, the precursor ion was 106.0506 and the fragment ion 60.0454 was used for extraction ion chromatogram. For $^{13}$C serine, the parent ion was 109.0606 and the fragment ion 62.0521 was used for extraction ion chromatogram. The extraction ion mass window was 10 ppm. The NCE was 20.

## Serine and homocysteine measurements

To confirm that mtDNA depletion from EtBr treatment could also give rise to increased serine and decreased homocysteine, we seeded T-REx-293 cells to 6-cm plates at $5 \times 10^5$ ea in 3 ml DMEM + 10% FBS, with and without 100 ng/ml EtBr, and collected spent media samples after 3 d incubation.

To measure total homocysteine (*i.e.*, both oxidized and reduced), 200 µl spent media was reduced (*Magera et al., 1999*) by addition of 40 µl DTT (0.5 M), vortexed, centrifuged at 2000×g for 1 min to remove any cell debris, and the supernatant incubated at room temperature for 15 min. 120 µl of the supernatant was extracted by adding 200 µl 0.1% formic acid in ACN, vortexing for 15 s, incubating on ice for 30 min, and centrifuging at 13000×g for 20 min. The resulting supernatant was analyzed by LC-MS for homocysteine abundance as above. Homocysteine signal was acquired in SIM mode.

To measure serine, 200 µl spent media was pelleted at 2000×g for 1 min to remove cell debris, and 100 µl the supernatant extracted by adding 900 µl 0.1% formic acid in 75% ACN: 25% methanol, vortexing for 15 s, incubating on ice for 30 min, and centrifuging at 13000×g for 20 min. The resulting supernatant was analyzed by LC-MS for serine abundance as above. Serine signal was acquired in full scan mode.

## CRISPR knockouts

For single SHMT knockouts, $6\times10^5$ T-Rex293 cells in 6-wells were transfected with 300 ng U6-sgRNA PCR product (*Ran et al., 2013*) and 1 µg pCas9_GFP (Addgene #44719) using Roche X-tremeGENE 9. We used the guide sequence CATCTGCAATCTTCCGTAGC for SHMT1 and GCAACCTCAC-GACCGGATCA for SHMT2. 2–3 d after transfection, single cells from the top 5% of GFP expressors were deposited by FACS into 96-wells containing 50% spent media and 50% fresh media, supplemented with 200 µM serine and 1:500 normocin (InvivoGen). Single cell colonies were analyzed for SHMT1 and SHMT2 expression by Western blotting; SHMT1 antibodies were Cell Signaling Tech (Danvers, MA) #12612, and SHMT2 antibodies were Thermo Scientific #PA5-32228. Double SHMT knockouts were generated likewise, except transfections were with 225 ng of each U6-sgRNA PCR product, and FACS sorted cells were grown in media further supplemented with 1x HT (100x solution from Thermo Fisher; 16 µM thymidine and 100 µM hypoxanthine final). We also repeated SHMT1 and SHMT2 single knockout generation with 1x HT supplementation, and none of the knockout lines so generated were hypoxanthine or formate auxotrophs.

## Serine scrambling

Untreated and EtBr-treated SHMT1 KO T-REx-293 cells were seeded in 35 mm plates at $10^6$ and allowed to attach for 1 d. They were then washed with warm PBS and treated with labeling media (DMEM with 400 uM $^{13}C_3$-serine, 10% dialyzed FBS) for 1 hr. Cells were then washed with ice-cold PBS, and metabolites extracted using 750 ul 80% acetonitrile: 20% water. $^{13}C_3$-serine and $^{13}C_1$-serine were quantitated using the same procedure as for serine synthesis determination.

In follow-up studies, we used 50 uM $^{13}C_3$-serine on wild type cells. T-REx-293, 293T, and HeLa cells were seeded at $10^6$; C2C12 cells at $2.5\times10^5$; MCH58 at $5\times10^5$.

## NAD+/NADH assays

Cellular NAD+ and NADH content were determined as previously described (*Sullivan et al., 2015*).

## Formate synthesis in isolated mitochondria

We used a hybrid cavitation chamber (*Kristián et al., 2006*) / homogenizer (*Mootha et al., 1997*) technique to disrupt cells, followed by differential centrifugation to isolate mitochondria (*Mootha et al., 1997*). All steps for mitochondrial isolation were performed at 4°C. Briefly, 2–4 confluent 15 cm plates of T-REx-293 cells were washed with PBS (room temp) and scraped into a conical tube. These were pelleted at 600g for 10 min, aspirated, and resuspended in 11 ml IB$_c$ (200 mM sucrose, 10 mM Tris/MOPS, 1 mM EGTA/Tris, Roche cOmplete protease inhibitor, pH 7.4) (*Frezza et al., 2007*). The cell suspension was pressurized to 800 psi with nitrogen in a pre-chilled cavitation chamber (Parr Instruments, Moline, IL) for 15 min, and rapidly decompressed into a Potter-Elvehjem homogenizer. The cell suspension was then homogenized with 5 strokes at 1000 rpm. This lysate was centrifuged at 600 g to remove nuclei and intact cells. Mitochondria in the supernatant were then pelleted at 8000 g, and the supertatant aspirated. Mitochondria were resuspended in IB$_c$, and the differential centrifugation procedure repeated. After the second round, mitochondria were suspended in 300–400 µl IB$_c$, and quantitated by BCA assay (Thermo Fisher). Mitochondria were then pelleted and resuspended to 5.71 mg/ml in experiment buffer EB (137 mM KCl, 2.5 mM

MgCl$_2$, 10 mM HEPES, 1 mg/ml BSA, pH 7.4). Formate production assays were performed in 105 µl of EB with 150 µg mitochondria, 3 mM Pi, 1 mM serine, and 1 mM ADP, at 37°C for 20 min. Formate production was stopped by centrifuging the mixture at 8000 g for 10 min (4°C) and removal of supernatant. Replicate wells of 40 ul supernatant were analyzed using a formate assay kit (Sigma), where one well did not contain enzyme and was used as a background NAD(P)H control. Antimycin and rotenone were used at 1 µM, and the corresponding vehicle control was 0.1% (w/v) DMSO. When drug treatment was used, mitochondria were incubated with drugs for 2 min at 37°C prior to addition of serine, Pi, and ADP.

## Measurement of remethylation using deuterated formate

T-REx-293 cells were seeded to 6 cm plates at $2 \times 10^6$ and allowed to attach and grow for 1 d. They were then washed with warm PBS and treated with serine-free DMEM supplemented with 3 mM D-formate (Sigma) and 10% dialyzed FBS. After 12 hr treatment, cells were aspirated, washed with ice-cold PBS, and metabolites extracted using 1 ml 80% methanol: 20% water. Deuterated serine and methionine were quantitated as described previously (*Mascanfroni et al., 2015*). Briefly, cell extracts (10 µL) were diluted using 90 µL of 74.9:24.9:0.2 vol/vol/vol acetonitrile/methanol/formic acid containing stable isotope-labeled internal standards (valine-d8, Isotec; and phenylalanine-d8, Cambridge Isotope Laboratories; Andover, MA). The samples were centrifuged (10 min, 9000 g, 4°C) and the supernatants were injected directly onto a 150 × 2 mm Atlantis HILIC column (Waters; Milford, MA). The column was eluted isocratically at a flow rate of 250 µl/min with 5% mobile phase A (10 mM ammonium formate and 0.1% formic acid in water) for 1 min followed by a linear gradient to 40% mobile phase B (acetonitrile with 0.1% formic acid) over 10 min. The electrospray ionization voltage was 3.5 kV and data were acquired using full scan analysis over m/z 70–800 at 70,000 resolution. LC-MS data were processed and visually inspected using TraceFinder 3.1 software (Thermo Fisher Scientific; Waltham, MA).

## Growth curves

For each experimental condition, T-Rex293 cells were seeded in triplicate, either at $1.5 \times 10^5$/well in 12-well plates or at $3 \times 10^5$/well in 6-well plates, in DMEM + 10% dialyzed FBS, and split at 2 d or 3d intervals. Ethidium bromide was applied at 100 ng/ml, antimycin at 500 nM, rotenone at 100 nM, and oligomycin at 500 nM. CCCP was used at 5 µM when alone and 3 µM to rescue oligomycin treatment. On each split, cells were trypsinized, counted, and re-seeded to the original density. We used the cell counts to compute growth curves. Growth rates shown in *Figure 5—figure supplement 1a* and *Figure 5—figure supplement 1c* were computed by performing least-squares line fits to individual log-transformed growth traces from days 2–6. We omitted growth between days 0 and 2 because the drug-induced growth effects did not appear fully developed before day 2. 293MSR cells were grown identically. U251 cells were grown in triplicate in 6-cm plates at $1.2 \times 10^5$ per plate, and rotenone treatment on U251 was at 50 nM. U251 cells were split at 2 d intervals, and the growth rates shown in *Figure 5—figure supplement 3* are from days 2–10 in the U251 growth curves and days 0–6 in the 293MSR growth curves.

## Yeast

To revive the *S. cerevisiae* BY4741 strain, cells were plated onto YPD (rich medium) plates from frozen glycerol stocks. After 2 days, cells were taken from plates, re-suspended into liquid YPD, and counted. Next, an appropriate amount of cells were taken to inoculate a 3 mL culture of SD +2% Dextrose (Sunrise Science, San Diego, CA) at $1 \times 10^6$ cells/ml. The resulting 3 mL culture was placed in a New Brunswick Scientific (Edison, NJ) model TC-7 roller drum on the fastest rotation until saturated (16 hr). The cells were then counted and diluted back to $1 \times 10^6$ cells/ml in SD-serine-adenine+2% glycerol+2% ethanol with 5 nm Antimycin A. For the serine and adenine additions, 1X corresponds to 85.6 mg/L and 21 mg/L, respectively. To monitor growth in each condition, 150 uL of culture was placed in the wells of a 96-well plate and growth curves were done using the Bio Tek (Winooski, VT) Synergy H1 multi-mode plate reader. The growth conditions were 30°C with continuous low shaking. OD$_{600}$ was measured every 15 min for 48 hr.

## Statistics

All significance values were computed using Student's *t*-test (two-tailed) unless otherwise specified. All error bars shown in figures denote standard errors of the mean. *n* values denote biological replicates, *i.e.* measurements performed on independent biological samples.

## Acknowledgements

We thank Josh Baughman, Hany Girgis, Sarah Calvo, Denis Titov, Eran Mick, and other members of the Mootha lab for helpful discussions and technical assistance. We thank the MGH CRM Flow Cytometry Core and the DFCI Microarray Core for technical services, and David Thorburn, Vipin Suri, and Michael Baym for helpful discussions. This work was supported by NIH grant R01DK081457. VKM is an Investigator of the Howard Hughes Medical Institute.

## Additional information

### Competing interests

AR: is a senior editor at eLife. VKM: is a founder and holds a financial interest in Raze Therapeutics, which is targeting one-carbon metabolism for cancer. The other authors declare that no competing interests exist.

### Funding

| Funder | Grant reference number | Author |
| --- | --- | --- |
| Howard Hughes Medical Institute | | Xiaoyan Robert Bao<br>Olga Goldberger<br>Jun Peng<br>Rohit Sharma<br>Dawn A Thompson<br>Scott B Vafai<br>Aviv Regev<br>Vamsi K Mootha |
| National Institutes of Health | R01DK081457 | Xiaoyan Robert Bao<br>Vamsi K Mootha |
| National Institutes of Health | 2R01CA119176 | Aviv Regev |
| National Institutes of Health | R01HL101930 | Fumito Ichinose |

The funders had no role in study design, data collection and interpretation, or the decision to submit the work for publication.

### Author contributions

XRB, VKM, Conception and design, Acquisition of data, Analysis and interpretation of data, Drafting or revising the article; S-EO, JP, RS, DAT, SBV, AGC, EM, CBC, Acquisition of data, Analysis and interpretation of data, Drafting or revising the article; OG, Acquisition of data, Drafting or revising the article; FI, WG, AR, SAC, Analysis and interpretation of data, Drafting or revising the article

### Author ORCIDs

Xiaoyan Robert Bao, http://orcid.org/0000-0001-7931-2944
Vamsi K Mootha, http://orcid.org/0000-0001-9924-642X

## Additional files

### Supplementary files

• Supplementary file 1. Metabolite profiling data, showing raw mass spectrometry counts as well as adjusted abundance ratios (see Materials and methods).

• Supplementary file 2. Analysis of microarray data. (a) 100 most upregulated genes. Filtered probesets (see Materials and methods) were sorted according to mean upregulation between days 1 and

10 of mtDNA depletion. They were then mapped to genes, and redundant probesets were removed leaving only the most strongly upregulated for each gene. (b) 100 most downregulated genes. (c) motifADE results. Shown are all motifs showing an adjusted *p* value of $10^{-3}$ or less.

• Supplementary file 3. Protein profiling data. (a) Full SILAC data, summarized by protein. (b-e) 100 most enriched and depleted proteins (nominal p<0.05, *z*-test) with 2d mtDNA depletion (M/L SILAC ratio) and 5d mtDNA depletion (H/L SILAC ratio). (f) GSEA results, using 'weighted' enrichment statistic on log fold changes. Shown are top 10 gene sets associated with enriched and depleted proteins with 2 d and 5 d mtDNA depletion.

### Major datasets

The following datasets were generated:

| Author(s) | Year | Dataset title | Dataset URL | Database, license, and accessibility information |
|---|---|---|---|---|
| Bao XR, Goldberger O, Mootha VK | 2014 | Gene expression response to mitochondrial DNA depletion | http://www.ncbi.nlm.nih.gov/geo/query/acc.cgi?acc=GSE55311 | Publicly available at the NCBI Gene Expression Omnibus (accession no: GSE55311). |
| Bao XR, Ong SE, Carr SA, Mootha VK | 2016 | Mitochondrial dysfunction remodels one-carbon metabolism in human cells | ftp://massive.ucsd.edu/MSV000079791 | Publicly available at the Mass spectrometry Interactive Virtual Environment (MassIVE, accession no: MSV000079791) |

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
