## [Decision Letter]

Thank you for submitting your work entitled "Mitochondrial dysfunction induces remodeling of 1C metabolism in human cells" for peer review at *eLife*. Your article has been favorably evaluated by a Senior editor and three reviewers, one of whom, Utpal Banerjee, is a member of our Board of Reviewing Editors.

The reviewers have discussed the reviews with one another and the Reviewing editor has drafted this decision to help you prepare a revised submission. The revisions needed are not extensive, and most can be addressed by changes/explanations added to the manuscript or minor added results. We wish to convey our high enthusiasm for the appropriateness of this paper for *eLife* and anticipate that you will address these issues and return us the manuscript for rapid publication.

Summary:

Bao et al. integrate metabolomics, proteomics and RNA profiling to identifypathways affected by mitochondrial stress. The primary stressor employed is a doxycyline inducible POLG dominant -negative mutant that depletes mitochondrial DNA. Upon doxycycline treatment, the metabolite profiling experiments identified commonly seen indicators of metabolic stress, such as an increase lactate or decrease in uridine. However, they also observed novel changes, including increases in serine and decreases in homocysteine. RNA profiling identified changes in genes that might be responsible for these changes. They determine that the transcripts of enzymes involved in serine biosynthesis increase upon doxycycline treatment, as do enzymes involved in transsulfuranation. The authors next identified ATF4 as a possible transcription factor that mediates the transcriptional response to mtDNA depletion, and determined that ATF4 responsive genes also changed in expression in response to other mitochondrial stresses. Proteomic profiling confirmed that core mitochondrial protein complexes decreased in abundance, and that enzymes involved in serine biosynthesis are increased.

In follow-up experiments, the authors determined the effect of mtDNA depletion on H2S generation, serine consumption, and explore mechanisms by which mitochondrial dysfunction might compromise 1C metabolism. They identify that mitochondrial stressors cause a serine dependency that can be rescued by 1C donors, such as formate and hypoxanthine.

The manuscript has many positive aspects. The coordinated global analysis in which metabolite, protein and RNA levels are determined in response to mtDNA depletion, appears expertly performed and have yielded valuable information. The integration of these different data-sets is particularly impressive. The results are likely to be clinically important in proposing that mitochondrial dysfunction impacts upon serine and homocysteine metabolism. Indeed, if true for the in vivo situation, this information could be used to develop novel therapeutic approaches.

Essential revisions:

1) The primary mitochondrial stress used in this manuscript is a doxycycline inducible method to deplete mtDNA. Numerous reports have demonstrated that doxycycline inhibits mitochondrial protein synthesis. The lab of Johan Auwerx has published work recently, including a study in Nature, showing that 1 ug/ml doxycycline (the dose used in this study to turn on POLGdn expression) substantially decreases COX1 levels and decreases oxygen consumption in hek293 cells. Furthermore, other reports have determined that doxycycline increases ATF4 levels, in a POLGdn-independent manner. This study would be strengthened if the effect of doxycycline-dependent mtDNA depletion and doxycycline-dependent inhibition of mitochondrial protein synthesis were differentiated. An appropriate control was performed for the transcription analysis. Within this data-set, it would be useful to compare which genes are doxycycline sensitive and which respond only to mtDNA depletion (i.e. do the levels of ATF4 responsive genes change upon exposure to doxycycline)?

2) An important control experiment would be to determine whether control T-rex-293 cells treated with doxycycline increase serine or decrease homocysteine levels. Another key control would be to determine whether doxycycline treated cells demonstrate serine dependency.

3) In Figure 1—figure supplement 1 mtDNA copy number seems very high. In the original paper describing this system, Jazyeri et al. determined a copy number of ~500 before mtDNA depletion in this system (compared to ~20, 000 here), which is more consistent with other analyses in mammalian cell lines. In this manuscript, the cells still have ~1000 copies of mtDNA per cell following depletion. Given that mtDNA depletion is the primary stressor used, it will be useful to re-assess whether the values in the Figure are correct, or whether a minor technical error has occurred and these values need to be re-normalized.

4) In Figure 7, the authors demonstrate clearly that cells exposed to the mitochondrial toxins rotenone and antimycin, grow more slowly in the absence of serine. These experiments were performed using an alternative supplier of DMEM (US Biological compared to Invitrogen), so as the media could be made without serine. For those looking to repeat these experiments, it will be useful to specify more clearly in the Methods the DMEM components in the US Biological derived media. Specifically, it is necessary to state whether the home-made media contains the same amount of pyruvate as the Invitrogen media (1 mM pyruvate). Many commercial DMEMs do not contain pyruvate (including the base US biological DMEM), and variation in pyruvate experiments will greatly affect the results of these experiments.

5) The great strength of this paper is using a range of profiling technologies to link mitochondrial dysfunction to changes in serine (and homocysteine) metabolism. The authors hypothesize that mitochondrial dysfunction impacts upon NAD+/ NADH levels and that this then perturbs mitochondrial 1C metabolism. This is a very plausible hypothesis, but it is difficult to test directly, and the follow up experiments do not appear to fully and unambiguously resolve how mitochondrial dysfunction increases serine levels, lead to serine dependency and fully clarify the role of 1C metabolism in these processes. To further explore the role of 1C metabolism a detailed study of the levels of mitochondrial 1C metabolites would have been useful, although it is technically difficult, so may be beyond the scope of this study. For example, as the authors note, in CFD caused by mitochondrial DNA depletion 5-methyltetrahydrofolate levels are substantially decreased – it would be useful to measure 5-mTHF levels in these cells. Similarly, the authors do not directly demonstrate that mitochondrial NAD+/NADH levels are altered.

6) A great advantage of this system is that the profiling experiments can be very carefully controlled. A disadvantage of this system is that these cells are tolerant of severe mitochondrial stress (i.e. they can be used to make viable proliferating cells which contain no mtDNA). In fully differentiated cells, such as neurons or myocytes, which mtDNA depletion seems to affect most in vivo, the relative importance of the energetic component of mitochondrial stress and these secondary metabolic pathways is likely to be different. Full assessment of the relevance of these findings (such serine dependency upon mitochondrial stress) to metabolic disease will require the use of an animal model, but is beyond the scope of this study. The Discussion would be strengthened if a description of the limitations of the in vitro model were included.

7) An issue that is central to this paper is how ATF-4 function is regulated by redox changes. Does alteration of the redox status of a cell in the absence of mitochondrial insult also show similar phenotype?

8) (Minor point) A change in the redox balance induces many other changes like changes in the activity of SIRTUINS: do they also play a role in this response pathway?

9) The loss of a cell's ability to utilize glucose through TCA metabolism results in hypoxic response in many model systems. Is there a relationship of the present study to hypoxic response that also occurs under the conditions of RC stress?

10) Please comment on why was cysteine not analyzed, but homocysteine apparently did not pose a problem?

11) Authors should detail methods and metadata, if necessary due to length restrictions, even in extended supplement material sections, but at least briefly in the Methods sections in a summarized form.

12) In the Discussion section, the question posed is "An important question is the degree to which our cell culture data are relevant to in vivo mitochondrial disease?" However, the subsequent sentence is unclear both in grammar and style, but also in its meaning, i.e. in how far the answer relates to the initial question:

"We note, however, that more upstream measures of 1C compromise appear also to be more robust."

The reviewer assumes the authors want to say "More upstream signals of 1C damage appear also to be more robust" but that's not answering the question on in vivo relevance…

13) (Minor point) The authors wrote (subsection “Integration of profiling data”) their metabolite analysis method was a "high-throughput" method. Why would high-throughput be needed here? Is this high throughput like in research chemical analysis (e.g. in combinatorial chemistry libraries), where easily 3,000 analyses per hour can be conducted?

If such throughput is not meant in this case, such words should not be used. I have the vague conception the authors want to highlight the idea that more than one target is analyzed per run, but that is not "throughput".

14) In the Discussion, the authors acknowledge there may be questions regarding generality of these findings in vivo. But no literature data was provided that may question the generality of the conclusions. For example, in "Comparison of proteomic and metabolomic profiles of mutants of the mitochondrial respiratory chain in *Caenorhabditis elegans*" by Morgan et al. serine does not appear to be increased in *C. elegans* with RC defects. There must be other metabolomics publications on mitochondrial defects in vivo, and it would be nice to have more discussion on this issue.

15) The manuscript is quite complex and difficult for a non-specialist to grasp the significance of the results. Perhaps reducing the use of abbreviations could help, but it could also be structured in a simpler manner. Figure 5, for example, it is hard to understand the color coding (mRNA changes in blue look more like metabolites).

[Editors' note: further revisions were requested prior to acceptance, as described below.]

Thank you for resubmitting your work entitled "Mitochondrial dysfunction remodels one-carbon metabolism in human cells" for further consideration at *eLife*. Your revised article has been evaluated by a Senior editor, and two reviewers, one of whom is a member of our Board of Reviewing Editors. The manuscript has been improved but a rather important issue has been brought to light from the data in the submitted Supplementary material. As you know, biological and technical replicates are critical for the analysis to be statistically meaningful.

Using n=2 data set cannot be considered acceptable for any analysis. Even using an n=3 is only common for molecular biology testing of presence versus absence of bands e.g. for RNA or proteins, showing that e.g. a knockout or transgenic gene was correctly expressed. Using an n=3 for molecular phenotyping (proteomics, metabolomics) is marginal and for sure 2 is unacceptable.

Metabolite profiling data are given in supplement 1 (metabolite data n=2 per condition, i.e. a total of 12 samples plus one baseline control).

Protein data are given as supplement 3 (worksheet 2, columns AR to BC, total number of samples =12, but it is hard to say how those samples relate to the study groups). Likely similar to the metabolite data, but no clear headers are given.

In the Methods section:

"We characterized protein changes (n = 1) for two different time points, plus a control". In difference to the metabolite profiling, internal controls (heavy isotope labeled reagents) to correct for mass spec differences, but of course, this would not correct for biological variability if this experiment is replicated.

For transcript data these are uploaded to GEO. The authors say they have a total of 18 samples for a time course with n=1 ("RNA profiling. RNA from 18 time points (n = 1 for each time point…").

The statistical prowess of the data presented needs to be significantly improved. This may be difficult to achieve in a reasonable time frame for revisions, in which case, you may wish to send the manuscript to another journal. However, since the reviewers and editors all consider this work to be valuable and important, we are willing to expedite a new submission of the manuscript with the proper statistically significant data added. If this is going to be the course of action, we ask that you get in touch with our editorial staff, providing a plan of action that can be pre-assessed by our reviewer.

Minor issue:

Reviewer 2 also would like to see more details of the experimental methods. Please do include this in Supplementary Material or Methods section as you please if you decide to submit a new version of the paper. Below, we summarize the reviewer's comments, and once again, it will be useful to let us know whether you intend to include all or some of these experimental details:

"The metabolite profiling section, as stated, is not following common reporting standards in science, let alone the recommendations of the Metabolomics Standards Initiative. The authors state "Media and cellular metabolite extract (n = 2 for each) were characterised using liquid chromatography / tandem mass spectrometry following the procedure in (Jain et al., 2012) for polar metabolites (including organic acids), and the procedure in (Rhee et al., 2011) for lipids.

It is not sufficient to point to papers in which methods may, or may not, be explained in detail. In order to understand, interpret, and possibly repeat analyses, methods require detailed information, for example: (a) use of internal standards, and the readout of these standards. (b) use of quality controls, including negative and positive controls, for measurements and for biological as well as technical reproducibility. (c) details on HILIC columns use, solvents in the LC, gradients, flow rates, injection volumes, reconstitution solvents of the sample prior to injection, and volume of the reconstituted samples. (d) details on the mass spectrometry methods used, including e.g. declustering potentials and electrospray voltages, temperature for desolvation, precursor ion windows for Q1, use of multiple MRM transitions per metabolite as qualifier ions to confirm that in a MRM, the correct metabolite was analyzed. (e) use of calibration curves to determine sensitivity of analyses (incl LOD and LOQ).

Moreover, the MSI recommends detailed information given for each metabolite in the [Supplementary-material SD1-data], second worksheet, namely: mass, and MRM, for each metabolite. A database identifier (e.g. PubChem ID or InChI key) for each metabolite. And the retention time, for each metabolite.

The authors may want to choose uploading result data, and raw data, to one of the public metabolomic databases e.g. the NIH database metabolomics workbench.org or the European counterpart, Metabolites. For other types of data, similar repositories exist."

---

## [Author Response]

1) The primary mitochondrial stress used in this manuscript is a doxycycline inducible method to deplete mtDNA. Numerous reports have demonstrated that doxycycline inhibits mitochondrial protein synthesis. The lab of Johan Auwerx has published work recently, including a study in Nature, showing that 1 ug/ml doxycycline (the dose used in this study to turn on POLGdn expression) substantially decreases COX1 levels and decreases oxygen consumption in hek293 cells. Furthermore, other reports have determined that doxycycline increases ATF4 levels, in a POLGdn-independent manner. This study would be strengthened if the effect of doxycycline-dependent mtDNA depletion and doxycycline-dependent inhibition of mitochondrial protein synthesis were differentiated. An appropriate control was performed for the transcription analysis. Within this data-set, it would be useful to compare which genes are doxycycline sensitive and which respond only to mtDNA depletion (i.e. do the levels of ATF4 responsive genes change upon exposure to doxycycline)?

We thank the reviewers for pointing out the important work by the Auwerx lab on this topic. We have performed the requested analysis of ATF4 activation in T-Rex-293 cells treated with doxycycline. Using motifADE, we observed a subtle but significant association between the ATF4 motif TGATGCAA and increased expression upon doxycycline treatment. However, the magnitude of this association is much stronger in cells expressing PolGdn: the δ median metric for the association was 0.213 (with nominal *P*≈5.5×10^-9^) for PolGdn-expressing cells, whereas it was 0.016 (with nominal *P*≈1.6×10^-2^) for the parent T-REx-293 cells. Indeed, if we perform motifADE analysis on the doxycycline control data and consider all possible 8-mers, the signal associated with TTGCATCA is not significant due to Bonferroni correction of the *p*-value. Individual ATF4 target genes also show much weaker – if any – activation with doxycycline treatment in the absence of POLGdn expression (Figure 3—figure supplement 2). We now include a discussion of this finding in the main text, and compare degrees of gene activation in the two different cell lines in the new Figure 3—figure supplement 2. Furthermore, we note that we confirmed ATF4 activation in cells treated with ethidium bromide (EtBr) to induce mtDNA depletion (see Figure 3). Therefore, while doxycycline alone might contribute some small part of the transcriptional changes that we observe, the majority of those changes can be attributed to PolGdn overexpression and the ensuing mtDNA depletion.

2) An important control experiment would be to determine whether control T-rex-293 cells treated with doxycycline increase serine or decrease homocysteine levels. Another key control would be to determine whether doxycycline treated cells demonstrate serine dependency.

We agree with the reviewers’ concern here. However, because the suspected mode of off-target doxycycline action also involves mitochondrial damage, changes in 1C metabolites or serine dependency induced by doxycycline treatment alone would not necessarily disprove our model. Instead, the salient question is whether other modes of mitochondrial damage also induce these metabolic and growth changes. We have included new data showing increased serine and decreased homocysteine levels in spent media when cells are treated with ethidium bromide to induce mtDNA depletion (Figure 2—figure supplement 3). Furthermore, we note that our serine dependency growth studies were also performed using EtBr. We hope these additions and clarifications satisfy the reviewers’ concerns that the effects we observe arise solely as a result of doxycycline toxicity.

3) In Figure 1—figure supplement 1 mtDNA copy number seems very high. In the original paper describing this system, Jazyeri et al. determined a copy number of ~500 before mtDNA depletion in this system (compared to ~20, 000 here), which is more consistent with other analyses in mammalian cell lines. In this manuscript, the cells still have ~1000 copies of mtDNA per cell following depletion. Given that mtDNA depletion is the primary stressor used, it will be useful to re-assess whether the values in the Figure are correct, or whether a minor technical error has occurred and these values need to be re-normalized.

We thank the reviewers for pointing out the discrepancy. We wish to note two issues here. (1) The mtDNA copy number of 400-500 determined by Jazayeri et al.was not per cell, but *per nuclear genome* (Jazayeri et al., 2003). Since 293 cells are considered pseudotriploid (Lin et al., 2014), this actually corresponds to 1200-1500 copies per cell. (2) Literature reports for mtDNA copy numbers in HEK293 cells appear to vary. For instance, another paper (Wanrooij et al., 2007) reports a per cell copy number of around 3000 for a very similar HEK293-derived cell line.

We have re-assessed the calibration of mitochondrial DNA copy numbers, and discovered two technical errors that inflated the mtDNA copy number. Correcting these yields approximately 7000 mtDNA per cell (untreated), which is in much closer proximity to the aforementioned previous reports. In addition, we note that the precise copy number is not salient to any of our conclusions, and that we confirm abrogation of mitochondrial translation using Western blotting (Figure 1) and oxygen consumption measurements (Figure 2—figure supplement 1). Therefore, we have adjusted Figure 1—figure supplement 1 to only show relative mtDNA copy numbers.

4) In Figure 7, the authors demonstrate clearly that cells exposed to the mitochondrial toxins rotenone and antimycin, grow more slowly in the absence of serine. These experiments were performed using an alternative supplier of DMEM (US Biological compared to Invitrogen), so as the media could be made without serine. For those looking to repeat these experiments, it will be useful to specify more clearly in the Methods the DMEM components in the US Biological derived media. Specifically, it is necessary to state whether the home-made media contains the same amount of pyruvate as the Invitrogen media (1 mM pyruvate). Many commercial DMEMs do not contain pyruvate (including the base US biological DMEM), and variation in pyruvate experiments will greatly affect the results of these experiments.

We specifically requested US Biological to include 1 mM pyruvate in the custom-synthesized DMEM. We have amended the Methods section to address it.

5) The great strength of this paper is using a range of profiling technologies to link mitochondrial dysfunction to changes in serine (and homocysteine) metabolism. The authors hypothesize that mitochondrial dysfunction impacts upon NAD+/ NADH levels and that this then perturbs mitochondrial 1C metabolism. This is a very plausible hypothesis, but it is difficult to test directly, and the follow up experiments do not appear to fully and unambiguously resolve how mitochondrial dysfunction increases serine levels, lead to serine dependency and fully clarify the role of 1C metabolism in these processes. To further explore the role of 1C metabolism a detailed study of the levels of mitochondrial 1C metabolites would have been useful, although it is technically difficult, so may be beyond the scope of this study. For example, as the authors note, in CFD caused by mitochondrial DNA depletion 5-methyltetrahydrofolate levels are substantially decreased – it would be useful to measure 5-mTHF levels in these cells. Similarly, the authors do not directly demonstrate that mitochondrial NAD+/NADH levels are altered.

We have measured total NAD+/NADH levels in cells, and shown that the ratio does indeed decrease with either EtBr or rotenone treatment. These data are shown in the new Figure 6—figure supplement 2. Because both drugs affect cellular physiology by their action on mitochondria, it is highly unlikely that they could affect the total cellular NAD+:NADH ratio without altering the mitochondrial ratio.

As the reviewers correctly note, a detailed study of mitochondrial 1C metabolites would be very valuable for the current story. We indeed set up mass spectrometry methods to measure 5-methyltetrahydrofolate (5-mTHF) and were able to detect commercially available standards. Unfortunately, we were not able to detect 5-mTHF in T-REx-293 cell extracts, likely because 5-mTHF is not abundant in our cells. This conclusion is supported by our observation of very low rates of remethylation of homocysteine using formate-derived 1C units (Figure 6—figure supplement 4) despite a high expression level of methionine synthetase in our cells, because 5-mTHF is a cofactor in that reaction.

6) A great advantage of this system is that the profiling experiments can be very carefully controlled. A disadvantage of this system is that these cells are tolerant of severe mitochondrial stress (i.e. they can be used to make viable proliferating cells which contain no mtDNA). In fully differentiated cells, such as neurons or myocytes, which mtDNA depletion seems to affect most in vivo, the relative importance of the energetic component of mitochondrial stress and these secondary metabolic pathways is likely to be different. Full assessment of the relevance of these findings (such serine dependency upon mitochondrial stress) to metabolic disease will require the use of an animal model, but is beyond the scope of this study. The Discussion would be strengthened if a description of the limitations of the in vitro model were included.

We have added a discussion of the limitations of using a cell culture system that is tolerant of severe mitochondrial stress.

7) An issue that is central to this paper is how ATF-4 function is regulated by redox changes. Does alteration of the redox status of a cell in the absence of mitochondrial insult also show similar phenotype?

The precise mechanism of ATF4 activation by mitochondrial dysfunction is a very interesting question. Indeed, while both rotenone and antimycin were used at saturating concentrations, rotenone was much less effective at inducing the ATF4 response than was antimycin. Therefore, we suspect that redox changes are not the only signals of mitochondrial dysfunction that are salient for ATF4 activation.

As to the specific request of reviewers: we induced cellular redox imbalance by treating cells with lactate, and observed subtle but significant activation of the ATF4 response (Figure 3—figure supplement 4). The magnitude of the effect is much lower than that elicited by either mtDNA depletion or antimycin treatment, so other effects might also contribute, but it does appear that redox imbalance itself might be able to trigger the ATF4 response. We thank the reviewer for this excellent suggestion.

The nature of the signals emanating from damaged mitochondria that activates ATF4 will be an important topic for future studies.

8) (Minor point) A change in the redox balance induces many other changes like changes in the activity of SIRTUINS: do they also play a role in this response pathway?

We did not explore the role of sirtuins in the present study.

9) The loss of a cell's ability to utilize glucose through TCA metabolism results in hypoxic response in many model systems. Is there a relationship of the present study to hypoxic response that also occurs under the conditions of RC stress?

The connection between respiratory chain dysfunction and the hypoxic response remains controversial. While impaired respiratory chain function can activate HIF1 via altered NAD+/NADH ratios (Gomes et al., 2013) or increased cytoplasmic succinate (Selak et al., 2005), it can also inhibit HIF1 accumulation during hypoxia (Chua et al., 2010).

In our data, with a few notable exceptions, we did not observe activation of known HIF1 target genes.

More generally, we note that a prominent effect of the hypoxia response is to upregulate glycolysis, which provides – among other things – the substrate from which serine is synthesized. Serine is an allosteric activator of PKM2 (Chaneton et al., 2012), which is transcriptionally activated by HIF1. Therefore, we hypothesize that in settings where both ATF4 and HIF1 are activated, they may enhance each other’s adaptive effects. Understanding the interplay between these two pathways will require further experimentation.

10) Please comment on why was cysteine not analyzed, but homocysteine apparently did not pose a problem?

For unknown reasons, the targeted LC-MS method used for our initial metabolite profiling did not yield quantifiable cysteine signals using sample quantities that were sufficient to measure other amino acids. While the method could measure cysteine in neat reference standard solutions, the signal intensity was much lower in magnitude compared to the majority of other amino acids – including homocysteine – measured at similar concentrations, and the signals in biological extracts were noisy. It remains uncertain whether the lack of sensitivity for cysteine was due to interferences from co-eluting metabolites, interferences with the multiple reaction monitoring (MRM) MS settings used to measure cysteine in the method (m/z 122/76), or some other chemical interference related to the work up or the instrument.

11) Authors should detail methods and metadata, if necessary due to length restrictions, even in extended supplement material sections, but at least briefly in the Methods sections in a summarized form.

As far as we understand, the *eLife* format does not generally include extended supplement materials sections. As recommended by the *eLife* Research Article format, we have included fully detailed methods in the Methods section (now renamed “Materials and methods”) that appears after the Discussion.

12) In the Discussion section, the question posed is "An important question is the degree to which our cell culture data are relevant to in vivo mitochondrial disease?" However, the subsequent sentence is unclear both in grammar and style, but also in its meaning, i.e. in how far the answer relates to the initial question:

"We note, however, that more upstream measures of 1C compromise appear also to be more robust."

The reviewer assumes the authors want to say "More upstream signals of 1C damage appear also to be more robust" but that's not answering the question on in vivo relevance…

We apologize for the awkward wording and paragraph organization here. Our point here was to initially caveat the generality of serine dependency upon respiratory chain inhibition, since we see it in so few cell lines, and then remark that more upstream measures of 1C compromise appears to be more general. We have reorganized and expanded the relevant Discussion paragraphs with this critique, as well as critiques #6 and #14, in mind.

13) (Minor point) The authors wrote (subsection “Integration of profiling data”) their metabolite analysis method was a "high-throughput" method. Why would high-throughput be needed here? Is this high throughput like in research chemical analysis (e.g. in combinatorial chemistry libraries), where easily 3,000 analyses per hour can be conducted?

If such throughput is not meant in this case, such words should not be used. I have the vague conception the authors want to highlight the idea that more than one target is analyzed per run, but that is not "throughput".

We agree – we have removed the term “high-throughput” from the manuscript.

*14) In the Discussion, the authors acknowledge there may be questions regarding generality of these findings in vivo. But no literature data was provided that may question the generality of the conclusions. For example, in "Comparison of proteomic and metabolomic profiles of mutants of the mitochondrial respiratory chain in Caenorhabditis elegans*" *by Morgan et al. serine does not appear to be increased in* C. elegans *with RC defects. There must be other metabolomics publications on mitochondrial defects in vivo, and it would be nice to have more discussion on this issue.*

We have included a discussion of in vivo datasets in the revised Discussion section.

15) The manuscript is quite complex and difficult for a non-specialist to grasp the significance of the results. Perhaps reducing the use of abbreviations could help, but it could also be structured in a simpler manner. Figure 5, for example, it is hard to understand the color coding (mRNA changes in blue look more like metabolites).

We apologize that the key for Figure 5 was confusing; we have modified it to make it less ambiguous. Furthermore, in the present revision, we have reorganized the Results and Discussion sections, and made numerous smaller changes throughout the paper, with the goal of making the paper easier to read.

[Editors' note: further revisions were requested prior to acceptance, as described below.]

The manuscript has been improved but a rather important issue has been brought to light from the data in the submitted Supplementary material. As you know, biological and technical replicates are critical for the analysis to be statistically meaningful.

Using n=2 data set cannot be considered acceptable for any analysis. Even using an n=3 is only common for molecular biology testing of presence versus absence of bands e.g. for RNA or proteins, showing that e.g. a knockout or transgenic gene was correctly expressed. Using an n=3 for molecular phenotyping (proteomics, metabolomics) is marginal and for sure 2 is unacceptable.

[…] The statistical prowess of the data presented needs to be significantly improved. This may be difficult to achieve in a reasonable time frame for revisions, in which case, you may wish to send the manuscript to another journal. However, since the reviewers and editors all consider this work to be valuable and important, we are willing to expedite a new submission of the manuscript with the proper statistically significant data added. If this is going to be the course of action, we ask that you get in touch with our editorial staff, providing a plan of action that can be pre-assessed by our reviewer.

Our revision was rejected on the grounds of insufficient replicates in the initial microarray, proteomic, and metabolomic profiling experiments. We agree that the individual profiling experiments were not performed with sufficient statistical power to justify any conclusions when considered in isolation. However, we performed these profiling experiments with the goal of generating hypotheses, and in fact, focused our follow-up on a pathway that emerged from all three profiling modalities. The title claim of the paper is supported by no fewer than ten different follow-up experiments, each with *n* = 3 or more.

Please consider the analogy to a paper in which a drug screen is performed – rarely will a drug screen be performed with a large number of replicates (typically, two replicates are performed) – but then the follow-up focuses on a few key hits. In our case, the RNA, protein, and metabolite profiles were performed as a “screen,” designed to generate hypotheses, which we then pursued.

We do understand the institutional prerogatives of *eLife* to uphold the highest standards of statistical rigor in the papers that it publishes, and to avoid setting precedents that compromise these standards. Therefore, we have further revised the paper to caveat the interpretation of the profiling results:

· We have modified the Abstract to emphasize the hypothesis-generating nature of our RNA, protein, and metabolite profiling experiments.

· We have removed individual profiling panels from the main figures and placed them into figure supplements, to de-emphasize them. Instead, the new Figure 1 shows a joint analysis of all the profiling modalities.

· In the text, we have removed discussion of results from individual profiling modalities.

· We have added an extra paragraph that emphasizes the use of the initial profiling data sets in generating, not validating, hypotheses. This paragraph furthermore cautions against interpreting the profiling data in isolation, because of the small *n* used.

· Where appropriate, we have emphasized the use of validating experiments to turn hypotheses raised by profiling results into well-founded conclusions.

Minor issue:

Reviewer 2 also would like to see more details of the experimental methods. Please do include this in Supplementary Material or Methods section as you please if you decide to submit a new version of the paper. […]

As requested by Reviewer 2, we have amended the Methods section relating to the initial metabolite profiling data set, and (where possible) have annotated [Supplementary-material SD1-data] with metabolite PubChem IDs and mass spectrometry parameters.